# MultiModal Representation Learning for MultiSensory Video Simulation

## Abstract

General-purpose household robots require real-time fine motor control to handle delicate tasks and urgent situations. In this work, we introduce the senses of proprioception, kinesthesia, force haptics, and muscle activation to capture such precise control. This comprehensive set of multimodal senses naturally enables fine-grained interactions that are difficult to simulate with unimodal or text conditioned generative models. To effectively simulate fine-grained multisensory actions, we develop a feature learning paradigm that aligns these modalities while preserving the unique information each modality provides. We further regularize action trajectory features to enhance causality for representing intricate interaction dynamics. Experiments show that incorporating multimodal senses improves simulation accuracy and reduces temporal drift. Extensive ablation studies and downstream applications demonstrate effectiveness and practicality of our work. [‡]

## 1 Introduction

For general-purpose household robots to operate dexterously and safely like humans, they need to be enabled with multipotent sensory systems. Our interoceptive senses, including kinesthesia, proprioception, force haptics, and muscle activation, work together to enable us to dynamically engage with our surroundings. The ability to simulate such multisensory actions is crucial for developing robust embodied intelligence and guiding future directions for sensor design.

Traditionally, physics engines are used to simulate state changes of the environment (Tian et al., 2022; Tang et al., 2023; Mendonca et al., 2021; Li et al., 2023a; Hansen-Estruch et al., 2022), but creating a physics simulator with fine-grained multisensory capabilities for diverse tasks is both computationally expensive and complex in engineering. Recent works (Yang et al., 2023; Du et al., 2023) demonstrate the potential to use text-conditioned video models as simulators, but text struggles to capture the delicate control needed for tasks such as culinary or surgical activities. In this work, we introduce multisensory interaction signals in generative simulation to enable fine-grained control.

We focus on learning an effective multimodal representation to control generative simulation. Prior works on multimodal feature learning (Girdhar et al., 2023; Zhu et al., 2023; Shah et al., 2023; Ilharco et al., 2021; Du et al., 2021; Li et al., 2023b) focus the task of cross-modal retrieval. They thus emphasize multimodal alignment but overlook the unique information each modality provides. As a result, they are insufficient for conditioning generative simulators. For our task, we introduce an multimodal feature extraction paradigm that align modalities to a shared representation space while preserving the unique aspects each modality contributes. Additionally, we propose a generic feature regularization scheme to ensure the encoded action trajectories to be more context-and-consequence-aware, allowing for seamless integration with downstream video generation frameworks.

In this work, we introduce multisensory interoceptive signals of haptic forces, muscle stimulation, hand poses, and body proprioception to generative simulation for fine-grained responses. We focus on learning effective multisensory action representation to control generative video models. Our proposed multimodal feature extraction paradigm aligns different sensory signals while preserving the unique contributions from each modality. Additionally, we introduce a novel feature regularization scheme that the extracted latent representations of action trajectories to capture the intricate causality in context and consequences in interaction dynamics. Extensive comparisons to existing methods

---

[‡]For further references: https://sites.google.com/view/iclrsubmissionmultisensorysim/home?authuser=1

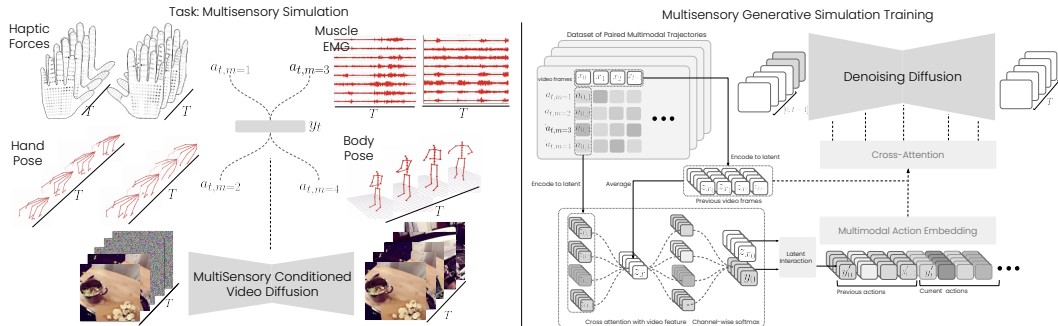

Figure 1: **Overview.** We introduce a new task to simulate fine-grained responses from multisensory interaction signals. We propose a generative simulation method, focusing on learning effective multimodal action representations to achieve fine-grained control of a video diffusion simulator.

shows that our multisensory method helps increase accuracy by 36 percent and improve temporal consistency by 16 percent. Ablation studies and downstream applications further demonstrate the effectiveness and practicality of our proposed approach. To summarize, our contributions are:

- To the best of our knowledge, we are the first to introduce multisensory signals, including touch, pose, and muscle response, to generative simulation for fine-grained responses.

- We devise a multimodal feature extraction paradigm that aligns modalities to a shared representation space while preserving the unique information each sensory modality provides.

- We propose a novel feature regularization scheme to enhance encoded action trajectories to be more context and consequence aware, capturing intricate interaction dynamics.

- We compare our proposed framework with prior approaches and also provide various possible downstream applications in policy optimization, planning, and more.

## 2 SIMULATING MULTI-SENSORY INTERACTIONS

We focus on two perspectives of modeling multi-sensory interactions. We first consider ways of working with **multi**modal signals, arriving at a multi-sensory action conditioning feature. We then focus on effective **inter**action modeling to capture the relationship between context and consequences in the learned representation. Finally, we cast our multisensory interoceptive action feature into a generative video model to simulate accurate exteroceptive visual responses.

**Problem Statement.** Simulators, at core, are next state prediction models. They estimate the consequential state changes of the world resulted from actions. Let $t \in [0, T]$ denote time frames, where $t \in [0, t-1]$ denotes the history horizon, and $t \in [t, T]$ are the future frames. For our task, at a snapshot of time $t$, we describe the state of the external world $s_t$ as visual observations $x_t \in \mathcal{O}$, that are the video frames and the set of sensory modalities denoted as $a_{t,m}$ of total number of $M$ modalities, $m \in [1, M]$. Given past observations $(\{a_{[0,t-1],m}\}, x_{[0,t-1]})$ and current action sequence $\{a_{[t,T],m}\}$, the goal of the simulator is to predict the consequential future states $s_{[t,T]}$ represented as a set of frames $x_{[t,T]}$. We denote the encoded video frame feature as $z_{x_t}$ that corresponds to $x_t | t \in [1, T]$, and we denote the encoded modality-specific features are denoted as $z_{t,m}$, and cross-modal feature is denoted as $y_t$. Under the generative simulation framework, we focus on extracting effective multimodal action representation $y_t$ from a set of multisensory actions $\{a_{[t,T],m}\}$ to condition a downstream generative simulator $g_\theta$ to accurately predict future states $x_{[t,T]}$.

### 2.1 MULTI-SENSORY ACTION REPRESENTATION

Multisensory actuation data are composed of temporal sequences of various sensory modalities of different granularity, dimension, and scale. How to effectively represent them, synchronize them, and combine them so they can accurately control a generative simulator are the three key challenges in generative *multimodal* feature learning.

One straight-forward way to extract feature representations from various sensory modalities is through mixture-of-expert (MoE) encodings. It is a commonly employed method for encoding heterogeneous data (Radevski et al., 2023; Mustafa et al., 2022; Riquelme et al., 2021). Various expert encoder heads $f_m(\cdot)$ are used to extract features $z_{t,m} = f_m(a_{t,m})$ that represent each sensory modality $m \in [1, M]$

at each time step $t$. To ensure that the encoded information in $z_{t,m}$ is meaningful, a self-supervised reconstruction scheme is introduced through MoE decoding branches $d_m(\cdot)$ across each sensory modality $\hat{a}_{t,m} = d_m(f_m(a_{t,m}))$ supervised by reconstruction loss, $\mathcal{L}_{\text{SSL}} = \|\hat{a}_{t,m} - a_{t,m}\|^2$, which gives rise to a set of MoE features $\{z_{t,m}\}_m^M$.

Before we combine these modality-specific features into a coherent multimodal feature, we need to synchronize them into the same representation space. Ideally, the synchronization strategy should align different MoE features to implicit follow some shared latent structure and simultaneously preserve uniqueness of each modality, *e.g.* hand pose can inform the action direction, while forces and muscle EMG both indicate action magnitude. These information should be meaningfully packed into different dimensions of the action feature. To encourage such association, we introduce an implicit cross-modal anchoring through channel-wise cross attention. We encode context video frames into latent vectors $z_{x_{[0,t-1]}}$ of dimension $d$, and obtain an anchor feature $z_{x_{\bar{t}}}$ by averaging across frames. We then use a learnable linear layer to project MoE features $z_{t,m}$ to anchor dimension $d$. Taking a channel-wise cross-attention between the anchor feature $z_{x_{\bar{t}}}$ and action features $\{z_{t,m}\}_m$ allows channels of the action latents $\{z_{t,m}\}_m$ to be associated through the channels of $z_{x_{\bar{t}}}$. In this way, we can train the linear projection layer to implicitly encourage a share latent structure to arise. Let $z_{t,m,j}$ denote the $j$-th dimension of the action latent vector $z_{t,m}$ of modality $m$ and timestep $t$.

$$z_{t,m,j} = \sum_i^d \frac{\exp z_{x_{\bar{t},i}} \cdot z_{t,m,j}}{\sum_{l=1}^d \exp z_{x_{\bar{t},i}} \cdot z_{t,m,l}} z_{t,m,j} \tag{1}$$

We are now ready to combine this set of modality-specific features $\{z_{t,m}\}_{m=1}^M$ into a cross-modal feature $y_t$. Different sensory modalities reflect different aspects of our actuation. These sensory modalities complement each other to provide comprehensive information about different actuations. This intuition suggests two properties of our multi-sensory input, over-completeness and permutation invariance. A good feature fusion function works as an information bottleneck to only select the most useful information. Moreover, unlike text sentences or image pixels, data of various sensory modalities is an unordered set. Therefore, the fusion scheme needs to be permutation-invariant regardless the modality order of the input. These properties encourage us to use symmetric functions for feature fusion. After comparing various symmetric functions (Sec. 3.3), we choose to use the softmax weighting function to aggregate different modalities of actuation,

$$y_t = \sum_{m=1}^M w_{t,m} z_{t,m}, \quad \text{where} \quad w_{t,m} = \frac{e^{z_{t,m}}}{\sum_{m'=1}^M e^{z_{t,m'}}}. \tag{2}$$

**Remark.** We avoid explicit alignment of the features through contrastive learning, as the task requires us to preserve differences between as some modalities that are *complementary*. The channel-wise softmax function helps us obtain a final vector allowing *substitutional modalities* to work together on the same dimensions. We observe that hand forces and the muscle EMG are highly correlated. In this way, these latent dimensions are implicitly attributed to reflect similar action property, *e.g.* strength for muscle and haptic forces, and thus increase robustness to missing modalities at test-time.

## 2.2 CONTEXT-AWARE LATENT REPRESENTATION OF INTERACTION

Previous steps have taken us to learn features that represent actions. Interaction is a special subset of action that bears the notion of contexts and consequences. We take one step further to investigate ways to represent **interaction**. An effective interaction feature should not only summarize the action property itself but engage with its contexts and hint at potential consequences.

**Latent Projection Interaction.** Under our task setting, interaction describes a way to take the observed context $x_{[0,t-1]}$ to the consequential states $x_{[t,T]}$. In the latent space, vectors that represent interactions are analogous to flow vectors that can be applied to various context states $z_{x_{[0,t-1]}}$ to the consequential changes states $z_{x_{[t,T]}}$.

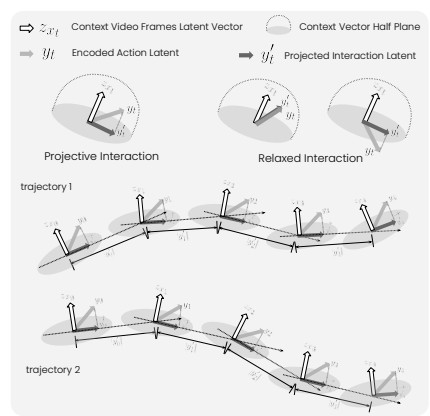

Figure 2: Latent Interaction

We wish to capture such effects in the latent vector itself.

Intuitively, the direction of latent interaction vectors $\{y_t'\}$ should consistently introduce similar effects relative to any context frames where they are applied. In other words, a good interaction vector should be locally constrained to its context frame, at the same time when applied to different contexts, the interaction vector should introduce similar behavior relative to the new context. These observations encourage us to constrain the behavior of action vectors through projective regularization. By removing the projected components on the context vector from the action vector, we extract the orthogonal component of the actions that reflects the dominant direction of change that an action can impose onto its context

$$y_t' = y_t - \left\langle y_t, \frac{z_{x_{t-1}}}{|z_{x_{t-1}}|} \right\rangle \frac{z_{x_{t-1}}}{|z_{x_{t-1}}|}. \tag{3}$$

In addition to direction constraint, we further capture the rate of such changes through an additional supervision signal, by matching the norm of the interaction vector $y_t'$ with the magnitude of frame-wise differences, $\mathcal{L}_{\text{NORM}} = \||y_t'| - |z_{x_t} - z_{x_{t-1}}|\|^2$. As shown in Fig. 2, these constraints help introduce the desired behavior in latent space. The two latent trajectories are formed by imposing the same interaction vector $y_t'$ to two different context frames $z_{x_0}$ and $z_{x_0'}$. Because the direction of change follows the orthogonal direction locally to the specific context frames and by the same magnitude, the two trajectories are similar.

**Relaxed Hyperplane Interaction.** A geometric interpretation of the latent interaction $y_t'$ reveals that the relative angle between context $x_{t-1}$ and interaction $y_t'$ depicts two spaces partitioned by a hyperplane defined by the normal vector $z_{x_{t-1}}$. This observation encourages us to rethink latent interaction modeling. The previous projection perspective forms a hard constraint where the interaction must follow the orthogonal direction of the context. In reality, interactions might induce slightly different behaviors when the context changes. Hence, we relax the hard orthogonal projection constraint. Through a geometric lens, the context vector $z_{x_{t-1}}$ can be viewed as a normal vector that defines a partitioning hyperplane, where interaction $y_t'$ with significant consequence to $x_{t-1}$ lies in the positive hemisphere, and negligible interaction resides below the hyperplane is clipped and projected.

$$y_t' = i(y_t, z_{x_{t-1}}) = \begin{cases} y_t & \text{if } \langle y_t, z_{x_{t-1}} \rangle \geq 0 \\ y_t - \left\langle y_t, \frac{z_{x_{t-1}}}{|z_{x_{t-1}}|} \right\rangle \frac{z_{x_{t-1}}}{|z_{x_{t-1}}|} & \text{otherwise} \end{cases} \tag{4}$$

We use this formulation to regularize interaction feature vectors $y'$ and adopt the magnitude constraint with frame-wise difference. The learned interaction feature $y_t'$ is used to condition the diffusion network pipeline to simulate future video frames.

## 2.3 Conditioning Generative Visual World Simulator

Inspired by (Yang et al., 2023; Ko et al., 2024), our simulator employs a video diffusion model to solve for future observations. Denoising video diffusion (Ho et al., 2020), in the forward process, predicts noise $\epsilon \sim \mathcal{N}(0, I)$ applied to the video frames $x_{[t,T]}$ according to a noise schedule $\bar{\alpha}^n \in \mathbb{R}$ over several steps $n \in [1, N]$, where $\bar{\alpha}^n = \Pi_{s=1}^n \alpha^s$. The optimization objective to train the video diffusion model $g_\theta$ is given by,

$$\mathcal{L}_{\text{VDM}} = \left\| \epsilon - g_\theta \left( \sqrt{\bar{\alpha}^n} x_{[t,T]} + \sqrt{1 - \bar{\alpha}^n} \epsilon, n \mid x_{t-1}, a \right) \right\|^2$$

For the task of future observation prediction, we use the learned model $g_\theta$ and reverse the process by iteratively denoising an initial noise sample $x_{[t,T]}^{n=N} \doteq \epsilon \sim \mathcal{N}(0, I)$ to recover video frames $x_{[t,T]}^{n-1}$ at denoising step $n - 1$. When $n = 0$, we obtain the estimated future video frames $\hat{x}_{[t,T]}$.

$$x_{[t,T]}^{n-1} = \frac{1}{\sqrt{\alpha_t}} \left( x_{[t,T]}^n - \frac{1 - \alpha^n}{\sqrt{1 - \bar{\alpha}^n}} g_\theta \left( x_{[t,T]}^n, n \mid x_{t-1}, a \right) \right) + \sigma, \sigma \sim \mathcal{N}(0, \frac{1 - \bar{\alpha}^{n-1}}{1 - \bar{\alpha}^n}(1 - \alpha)I)$$

We use I2VGen (Zhang et al., 2023) as our diffusion backbone. It uses a 3D UNet (Wang et al., 2023) with dual condition architecture that generates future video frames $x_{[t,T]}$ based on text prompt $a$ and context image $x_{t-1}$. We modify I2VGen (Zhang et al., 2023) replacing the single context frame with a history horizon of $h$ context frames by concatenating in the channel dimension. We also replace the text conditioning with our learned multimodal action feature $y_t$, where the cross

Figure 3: Comparison with unimodal conditioning.

| Method | MSE ↓ | PSNR ↑ | LPIPS ↓ | FVD ↓ |
|---|---|---|---|---|
| UniSim `verb` | 0.131 | 14.1 | 0.332 | 337.9 |
| UniSim `phrase` | 0.118 | 14.6 | 0.321 | 275.9 |
| UniSim `sentence` | 0.117 | 14.6 | 0.317 | 251.7 |
| Body-pose only | 0.127 | 14.4 | 0.345 | 295.9 |
| Hand-pose only | 0.122 | 14.5 | 0.349 | 307.6 |
| Muscle-EMG only | 0.134 | 13.8 | 0.364 | 348.2 |
| Hand-force only | 0.120 | 14.5 | 0.334 | 278.9 |
| Ours multisensory | **0.110** | **16.0** | 0.276 | 203.5 |
| Ours w/ `phrase` | 0.113 | **16.0** | **0.274** | **200.4** |
| Ours w/ `sentence` | 0.111 | **16.0** | **0.274** | 201.7 |

(a) Quantitative comparison

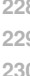

(b) Temporal drift. LPIPS per frame, learned perceptual image patch similarity.

attention is applied between noise frame samples and our conditioning feature $y_t$. Different from text-prompted simulation (Zhang et al., 2023; Yang et al., 2023), where a single text prompt $a$ is repeatedly used for all frames, our action condition is temporal, allowing our temporal attention to be frame-specific. (moved from end of sec. 2.2) We train the model end-to-end using a weighted sum of the aforementioned loss functions. The final supervision signal is given by $\mathcal{L} = \lambda_1 \mathcal{L}_{\text{VDM}} + \lambda_2 \mathcal{L}_{\text{SSL}} + \lambda_3 \mathcal{L}_{\text{NORM}}$, where $\lambda_1 = 10.0, \lambda_2 = 1.0, \lambda_3 = 0.1$. The relative weighting between different loss components $\{\lambda\}$ are chosen to align the magnitude of each component to the same level. We provide the details of our network architecture in Appendix Sec. 6.5.

## 3 EXPERIMENTS

We design our experiments to answer the following questions:

- Do we need multisensory action data to achieve fine-grained control over simulated videos?

- How do our multimodal feature extraction compare with existing ones when used for conditioning?

- Is our method robust to missing modalities at test time and how they influence prediction?

**Experimental Setup.** We use the ActionSense (DelPreto et al., 2022) dataset for our experiments as it is the first multi-sensory dataset with paired actuation monitoring and video sequences. The dataset collects five different interoceptions, including hand haptic forces, EMG muscle activities, hand pose, body pose, and gaze tracking. We use data recorded on subject five as our test set, and the remaining four subjects as training and validation set. We parse the dataset into paired sequences of 12 frames. We use the first four frame as the context frame and predict the remaining 8 frames. All experiments and methods use the same diffusion backbone, modified I2VGen (Zhang et al., 2023) (Sec. 2.3), which is a dual condition architecture that predicts video frames $x_{[t,T]}$ based on conditioning prompt $a$ and context image $x_{[0,t-1]}$. We vary the conditioning type $a$ for all experiments. All methods are trained from scratch on the same data with the same hardware and software setup. Due to computational constraints, our experiments are conducted with videos of $64 \times 64$ resolution. More details are included in Sec. 6.5

**Evaluation Metric.** We are interested in how various types of data and method used for conditioning can have different effects when simulating videos. We use the same video diffusion backbone, and vary the type and method for conditioning to observe the difference in simulated videos. We evaluate on a withheld test set from ActionSense (DelPreto et al., 2022), and use three different metrics to evaluate the quality of predicted video trajectories and the ground truth video trajectories, following (Yang et al., 2023). We use MSE, PSNR, LPIPS, and FVD scores as evaluation metrics to quantify the quality and accuracy of predicted video frames. In all tables, ↓ means lower is better for the metric, and ↑ indicates higher is better.

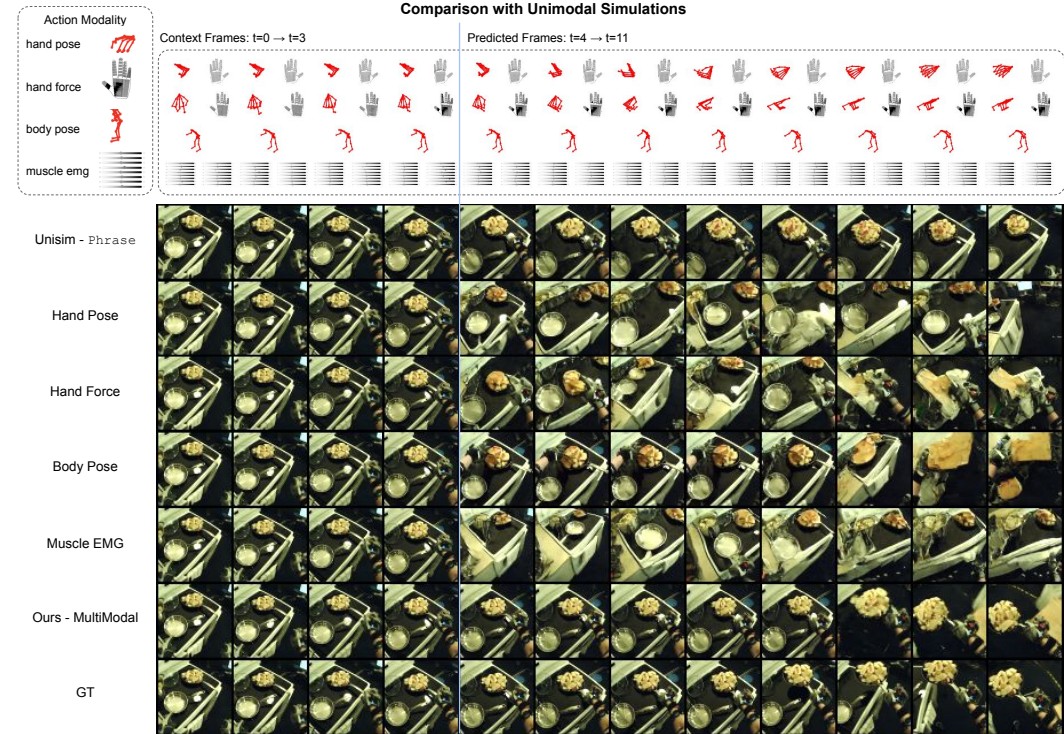

Figure 4: **Comparison to Unimodal Simulation.** We compare our proposed multisensory conditioning to unimodal conditioning, including text and each of the action sensory modality. The first four frames are the context history frames, and the last eight frames are predictions from each method.

## 3.1 ACTION CONDITIONING THROUGH TEXT, UNI-MODAL, MULTI-MODAL INPUTS

We are interested in understanding whether we need multisensory action data to achieve fine-grained control over simulated videos. To answer this question, we investigate the benefit of different action signal modalities, including text description, unimodal action, and multisensory action as input. For fairness of comparison, we use the same video generation model while varying the condition type.

**Comparison with Text-conditioned Simulation.** We first compare our proposed method and the state-of-the-art text-based video-diffusion simulator, UniSim (Yang et al., 2023). We vary the input condition with increasing details in description, using `verb`, `phrase`, `sentence`. `Phrase` are composed of verbs and subjects, *e.g.* `cut potato`. We add more detailed descriptions to form sentences, *e.g.* `person cut potato in a very fast manner, while holding it with left hand`. As shown in Table. 3a, our proposed method can achieve more accurate future frame prediction, significantly decrease temporal drift. Our method can take temporally fine-grained action trajectories with subtle differences as inputs to control the video prediction to match the action signals for each time step, whereas these subtle differences in the action trajectory are difficult to be accurately captured through text descriptions.

We show additional qualitative comparison in Fig. 8. We can see from the figure that our proposed method can be used to generate more diverse video trajectories from the same context frames, whereas other text-conditioned video simulation is more prone to mode collapse, converging to similar future video frames when given similar context frames. These new video trajectories generated with our method can be used for data augmentation to compensate the scarcity of paired action video data. As shown in Table. 3a and Fig. 8, adding `text phrase` as an additional modality to our method can help reduce model confusion. Additional discussion is included in Appendix Sec. 6.7.1.

**Comparison with Unimodal Action Simulation.** While text lacks the temporally fine-grained property, we extend our experiments to test the necessity of **multi**modal interaction by comparing to each of the action modalities alone. As there lacks direct baseline method that utilizes these action modalities for simulation, we use our own method for encoding these modalities and conditioning

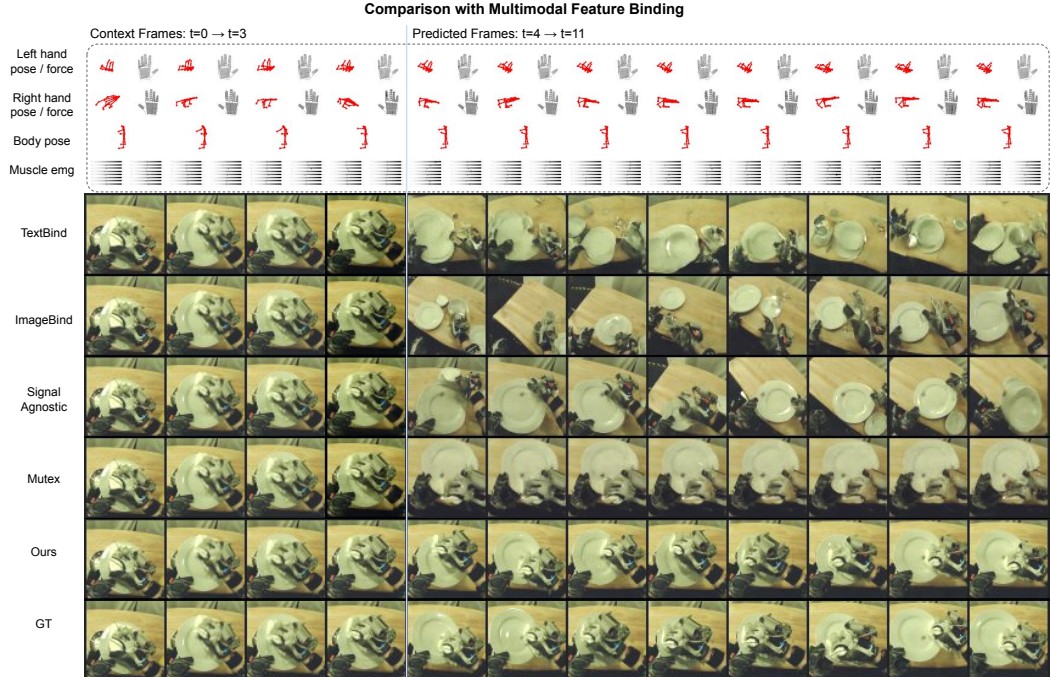

Figure 5: Comparison with multimodal feature extraction baselines

video models. The closest work to one of our unimodal baseline setting is Karras et al. (2023a), which uses a two stage finetuning of stable diffusion to generate full-body videos from pixel-level dense poses assuming static camera. The assumptions of dense poses, static camera, and full-body video make it difficult and unfair for this method to tackle our task setting with egocentric videos.

The middle section in Table. 3a shows that future video frame prediction is most accurate when all modalities are combined together. This is because not all modalities are created equal, and our ability to swiftly control and operate with our surroundings is a multiplicative effect of different functions working together. As shown in Fig. 4, a simple task of removing the pan from the stove top requires us to reach to the pan (body pose), grab the pan (hand pose and force), lift the pan (muscle and body pose), and finally turn around(body pose). When only training with hand-forces, the model has no information to locate the hand with respect to the environment, and thus generate hand holding random things in the image instead of the pan and results drift off (Fig. 4). We almost never entirely isolate one sense to interact with the world. Therefore, training with a single modality is not enough for such tasks, even when each signal is temporally fine-grained.

## 3.2 MULTI-SENSORY FEATURE EXTRACTION FOR GENERATIVE SIMULATION

For the task of multisensory action controlled simulation, we compare our proposed multisensory feature learning scheme to existing approaches to see how they induce the effects of interaction in explicit pixel space. We compare our method with various state-of-the-art multimodal feature extraction paradigm (Girdhar et al., 2023; Zhu et al., 2023; Shah et al., 2023; Du et al., 2021):

Figure 6: Comparision on multimodal feature extraction for generative simulation.

| Method | MSE ↓ | PSNR ↑ | LPIPS ↓ | FVD ↓ |
|---|---|---|---|---|
| Mutex | 0.164 | 12.4 | 0.431 | 410.1 |
| Imagebind | 0.134 | 13.9 | 0.390 | 315.6 |
| Languagebind | 0.143 | 13.7 | 0.387 | 332.0 |
| SignalAgnostic | 0.127 | 14.3 | 0.361 | 267.5 |
| Ours | **0.110** | **16.0** | **0.276** | **203.5** |

(a) Quantitative comparison

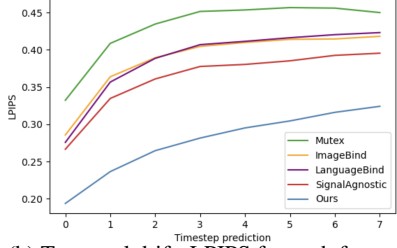

(b) Temporal drift. LPIPS for each frame.

- Mutex (Shah et al., 2023) proposes to randomly mask out and project some of the input modalities and directly align and match the remaining modalities to future frames.

- LanguageBind (Zhu et al., 2023) proposes to use text as a binding modality instead of using images.

- ImageBind (Girdhar et al., 2023) is a contrastive binding technique that leverages InfoNCE (Oord et al., 2018) contrastive loss to bind different modality of features to clip-encoded image features.

- Signal-Agnostic learning (Du et al., 2021; Li et al., 2023b) extracts cross-modal feature using signal-agnostic neural field.

As shown in Table. 6a, our proposed multi-sensory interaction feature outperforms other baseline method for multi-modal feature extraction for the task controlled generative simulation. Different multimodal tasks demand different representational properties. Previous approaches to multimodal feature learning (Girdhar et al., 2023; Zhu et al., 2023; Ruan et al., 2023; Lyu et al., 2023; Radford et al., 2021) are proposed for the task of cross-modal retrieval, emphasizing the interchangeability between modalities by extracting shared information through contrastive learning or modality anchoring. However, in the context of generative simulation, each action modality captures unique aspects of human behavior; they are both substitutional and complementary. Specifically, TextBind (Zhu et al., 2023) use contrastive loss to align various signal modalities to the encoded text descriptions. Constrastive losses magnify similarity between the participating features. Thus, training to match action sensory features to text features wipes out the temporal fine-grained information from the encoded action signals, leading to compromised predictions. Similarly, ImageBind (Girdhar et al., 2023) and Mutex (Shah et al., 2023) aligns action signal modalities to the encoded video frames, where Imagebind (Girdhar et al., 2023) uses contrastive loss to align action and visual features and Mutex (Shah et al., 2023) uses L2 loss to directly regress the features between various modalities and the pretrained CLIP encoded visual feature. As very similar action motion trajectories can work with different visual contexts, matching action modality feature directly to various visual context creates a one-to-many mapping problem, making it difficult for the network to extract the intrinsic motion from the visual context, leading to significant error accumulation. Moreover, action signals and visual observation are modalities of large spatial disparity, directly regressing them leading to mode collapse when predicting future video frames. Signal Agnostic Learning (Du et al., 2021; Li et al., 2023b) on the other hand does not use contrastive learning. By allowing gradient from different signal modalities to directly optimize the same latent manifold, Signal Agnostic approaches seem to outperform other baseline methods. However, these approaches induce loose coupling between the action signal modalities and the exteroceptive video modality, resulting in significant error accumulation.

As a result, generative simulation requires a distinct representation strategy that preserves this dual nature. To meet these requirements, our propose feature extraction scheme is better suited for this task.

### 3.3 ABLATION EXPERIMENTS

We provide three sets of ablation experiments to study how different senses help with simulation. We also conduct ablation studies to validate various design choices and effect of history horizon length.

**Interoceptive Sensories.** We first ablate different sensory signal input, when training our video simulator. We observe that body pose is crucial for larger motions that involve moving in space such as turning or walking. For more delicate manipulations such as cutting or peeling, hand poses and haptic forces get us most of the way. Results in Table 1a suggests that contribution of muscle EMG is minimal. A closer look into the dataset reveals that muscle EMG is highly correlated with mean hand force magnitude, but it provides extra information in scenarios where hands are fully engaged.

**Robustness to Missing Modalities during Test Time.** We are interested in understanding the extend of our test-time robustness to missing modalities. We evaluate our model trained on all modalities with each of the modalities removed, shown in Table 1b. We can see that the prediction accuracy of our model is slightly influenced by ablated modalities during test time. From the right side of Fig. 7, we can see that our model can still make sensible predictions under missing modalities, although prediction is most accurate with all modalities included. The left side of the Fig. 7 shows a stress test evaluating our model provided with only one modality. We see when that the hand pose trajectory is more accurate compared to other ones, which hint at a task-specific critical modality. More details on the stress test can be found in Appendix 6.7.2.

Figure 7: **Robustness to missing modalities during test time.** Left side shows stress test with evaluating with one single modality *provided*. Right side shows testing with one modality *removed*. For clearer visualization, we show the last context frame $x_{t-1}$ and the predicted video frames $x_{[t,T]}$.

**Comparison between Training and Testing with Ablated Modalities** The critical difference between the above two experiments, training with ablated modalities (Table. 1a) and testing with missing modalities (Table. 1b) is the modalities used during training. The latter ablation experiment, testing with missing modalities, employs a model trained with all modalities, whereas the former is trained only on a subset of modalities. Comparing the performance decrease in Table. 1a and Table. 1b, we can see that the latter experiment, testing with missing modalities, induces very minimal drop in prediction accuracy. This comparison confirms the advantage of training on multimodal action signals. We believe that this test-time robustness is induced by channel-wise attention and channel-wise softmax module, as these design choices allows the model to leverage substitutional information in the given modalities to bridge different modalities to allow for robustness during inference.

**Multimodal Feature Extraction** We are interested in understanding how various multi-sensory feature fusion strategies result in differences in simulated video trajectories. We compare with commonly employed symmetric functions for multi-modal fusion to validate the performance of softmax-ensemble approach. We can see from Table 1d that softmax outperforms mean pooling and max pooling. We refrain from using direct feature concatenation to preserve the permutation invariance property of the multi-sensory data. Direct concatenation is less robust when some sensory modalities are unavailable during test-time.

Additionally, we show an ablation experiment to validate our interaction feature $y'$ learning scheme. We can see from Table. 1d that when removing interaction module and directly using action feature $y$ as condition, the performance drops drastically. Action feature contains all information about the action itself, but not all information is meaningful to change the context frame. Action features are

Table 1: Ablation Experiments on Sensory Modalities and Network Components

| Method | MSE ↓ | PSNR ↑ | LPIPS ↓ | FVD ↓ |
|---|---|---|---|---|
| No hand pose | 0.138 | 14.1 | 0.314 | 264.0 |
| No hand force | 0.129 | 14.5 | 0.317 | 256.3 |
| No body pose | 0.137 | 14.5 | 0.322 | 273.1 |
| No muscle EMG | 0.121 | 15.2 | 0.311 | 217.1 |
| All sensory used | **0.110** | **16.0** | **0.276** | **203.5** |

(a) Training with ablated modalities

| Method | MSE ↓ | PSNR ↑ | LPIPS ↓ | FVD ↓ |
|---|---|---|---|---|
| No hand pose | 0.111 | 15.3 | 0.304 | 205.1 |
| No hand force | 0.113 | 15.5 | 0.307 | 205.0 |
| No body pose | 0.115 | 15.3 | 0.304 | 205.6 |
| No muscle EMG | 0.113 | 15.2 | 0.291 | 204.7 |
| All sensory used | **0.110** | **16.0** | **0.276** | **203.5** |

(b) Testing with missing modalities

| Method | MSE ↓ | PSNR ↑ | LPIPS ↓ | FVD ↓ |
|---|---|---|---|---|
| Unisim h(x) = 1 | 0.177 | 12.7 | 0.408 | 674.9 |
| Unisim h(x) = 4 | 0.118 | 14.6 | 0.321 | 275.9 |
| Ours h(x) = 1 | 0.142 | 12.9 | 0.362 | 535.1 |
| Ours h(x, a) = 1 | 0.138 | 12.7 | 0.356 | 529.1 |
| Ours h(x) = 4 | 0.114 | 15.4 | 0.306 | 256.3 |
| Ours h(x, $a_h$) = 4 | **0.110** | **16.0** | **0.276** | **203.5** |

(c) Effects of history horizon length

| Method | MSE ↓ | PSNR ↑ | LPIPS ↓ | FVD ↓ |
|---|---|---|---|---|
| Max | 0.128 | 14.1 | 0.294 | 284.8 |
| Mean | 0.126 | 14.4 | 0.293 | 285.3 |
| Concatenation | 0.117 | 15.0 | 0.282 | 279.9 |
| Without $y'$ | 0.142 | 13.7 | 0.327 | 339.0 |
| Projection $y'$ | 0.116 | 14.5 | 0.288 | 265.5 |
| Ours full | **0.110** | **16.0** | **0.276** | **203.5** |

(d) Ablation of network components or alternatives

less effective when the downstream video model fail to capture the right information to focus on, and consequently result in mode collapse. When we add the hard projection regularization $y'$, the accuracy of predicted video significantly improves, but still marginally worse compared to our full pipeline, which uses the relax hyperplane interaction scheme.

**History Horizon.** Finally, we study the effect of history horizon length on our model with comparison to text-conditioned simulation. We follow prior works (Yang et al., 2023) to compare context frame length h(x)=4 and h(x)=1, shown in Table 1c. We can see that increased history frame length reduces prediction error for all methods. Additionally, our proposed multisensory action condition is temporally fine-grained, which allows the cross attention between action and observation history h(x, a) = 4 to help further increase simulation accuracy.

## 4 DOWNSTREAM APPLICATIONS

**Low-level Policy Optimization** One downstream application of our proposed action-conditioned video generative simulator is to optimize a policy of low-level actuation. Inspired by (Yang et al., 2023), We set up task as goal-conditioned policy optimization, where we optimize a policy to generate a trajectory of low-level actuation $a_{[1,T]}$ that brings the environment from start state $s_0$ to target $s_T$. States are described by images $s_t \doteq x_t$.

We show one use case of our model in goal-conditioned policy optimization. We compare training of the same policy network $p(\cdot)_{\pi_\theta}$ under two conditions. **First**, we define the baseline method using the commonly employed goal-conditioned policy training approach (Reuss et al., 2023; Ding et al., 2019; Chi et al., 2023b). This baseline is the policy network taking the starting state and target state, depicted by two video frames $x_0$ and $x_T$, and directly regress policy $\pi_\theta$ minimizing the L2 distance between the predicted action $\hat{a}[1,T] = \pi_\theta(x_0, x_T)$ and ground truth expert action trajectory $a_{[1,T]}$. This L2 loss term is defined as $\mathcal{L}_a = \|\sum_t \hat{a}_t - a_t\|_2 = \|p(x_0, x_T)_{\pi_\theta} - a_{[1,T]}\|_2$. The **second** condition is to train the same policy $\pi_\theta$ in conjunction with our pretrained simulator. We feed the action trajectory predicted by policy network $\hat{a}_{[1,T]} = \pi_\theta(x_0, x_T)$ into our pretrained simulator model $g(\cdot)$ to predict the video frames from this action trajectory $\hat{x}_T = g(p(x_0, x_T)_{\pi_\theta})_T$. This additional loss term is defined as $\mathcal{L}_{sim} = \|\hat{x}_T - x_T\|_2 = \|g(p(x_0, x_T)_{\pi_\theta})_T - x_T\|_2$. The total loss term for the second condition is $\mathcal{L}_{simpolicy} = \mathcal{L}_a + \mathcal{L}_{sim}$. We evaluate the effectiveness of by using L2 distance between the predicted action $\hat{a}_{[1,T]}$ and ground truth action $a_{[1,T]}$, which is defined $\|\hat{a}_{[1,T]} - a_{[1,T]}\|_2$. (replace the original version of this paragraph:) We use our generative simulator model $g(\cdot)$ trained on real-world videos to simulate videos from the action outputs $a_{[\hat{1},T]}$ produced by policy network $p(\cdot)$. We use MSE loss between the last frame of the simulated video $g(p(x_0, x_T)_{\pi_\theta})_T$ and goal state $x_T$, $\|g(p(x_0, x_T)_{\pi_\theta})_T - x_T\|_2$ as an additional supervision signal to optimize the policy. Specifically, we use diffusion policy (DP) (Chi et al., 2023a) as the policy network $p(\cdot)_{\pi_\theta}$ to optimize $\pi_\theta$ that goes start state $s_0$ to end state $s_T$. We compare the performance of policies trained with and without our simulator. We show policy MSE which is the L2 distance of action trajectories of the optimized policies and the true action trajectory. $x_T$ MSE is a supporting

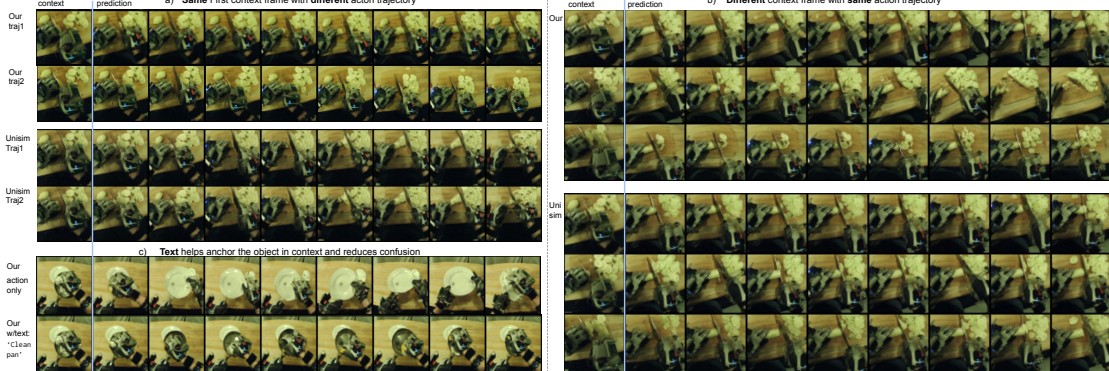

Figure 8: **Simulating new video trajectories** Comparing our multisensory method and text-based Unisim in generating diverse video trajectories from same or different context frames. For clearer visualization, we show the last context frame $x_{t-1}$ and the predicted video frames $x_{[t,T]}$.

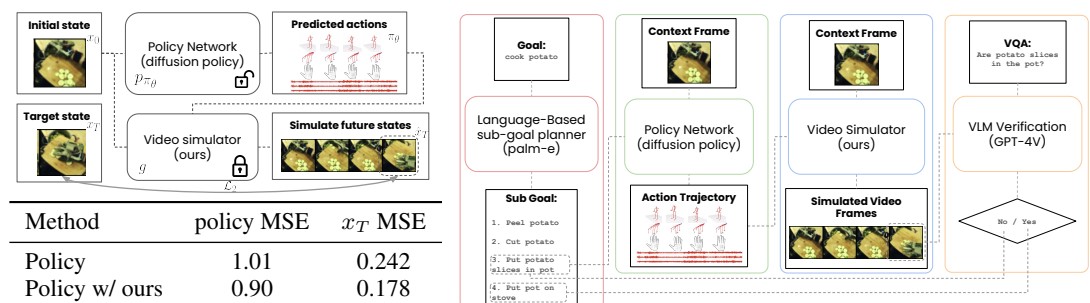

| Method | policy MSE | $x_T$ MSE |
|---|---|---|
| Policy | 1.01 | 0.242 |
| Policy w/ ours | 0.90 | 0.178 |

Figure 9: **Left:** Pipeline for goal-conditioned policy optimization. **Right:** Pipeline for long-term task planning.

metric that compares target state and the simulated end state using our simulator. Unfortunately, there is no other simulator for multisensory actions of such that we can use for further validation.

We can see from our experiments in Fig. 10 that adding our additional supervision signal helps to improve policy optimization. Directly regressing multi-sensory actions with a policy network is difficult because the action space in our task setting is quite large. The multi-sensory action space is 2292 dimensional. Additionally, we also observe that the policy optimized by our proposed approach can be different from the ground truth action trajectories, yet the simulated visual observations still closely resemble the ground truth state observations. We believe that the softmax aggregation learns to pick out information deemed useful by the simulator, leaving freedom in irrelevant dimensions in the action space. More results are included in Appendix Sec. 6.7.

**Multi-Sensory Action Planning** Another potential downstream application is long-term planning. Inspired by (Du et al., 2023), we use text to describe high-level goals to generate a set of executable next-step actions. Our video model takes an image observation and the generated actions to simulate future image sequences, which can be further evaluated for next-step execution planning. As shown in Fig. 9, our model can potentially be used for low-level actuation planning through iterative action roll outs. We adapt diffusion policy (DP) (Chi et al., 2023a) to take in both first frame image feature $x_0$ and high-level goal $\gamma$ described by a text feature $f_\gamma$ as the context conditions to generate multi-sensory trajectories of fine-grained actions $a_{[1,T]} = p(x_0, f_\gamma)$. The action steps are then fed into our action-conditioned video generative model $g(\cdot)$ to generate sequences of future video frames $\hat{x}_{[1,t]} = g(x_0, a_{[1,t]})$. To decide whether the subtask $\tau$ has been achieved, we use a vision language model $f_v(\cdot)$ as a heuristic function (OpenAI & et al., 2024), which can be promted with the end state of the current roll out $\hat{x}_t$ to evaluate whether subgoal $\tau$ has been achieved. If more steps are needed, we can further iterate the process $a_{[t,it]} = p(\hat{x}_t, \gamma)$, $x_{[t,it]} = g(\hat{x}_t, a_{[t,it]})$. A sample result from text-promted diffusion policy is shown in Figure. 10. We observe long iterations result in accumulative error, as shown in the bottom row of Fig. 15 in Appendix Sec. 6.7). A larger-scale dataset can further boost performance for this task. This downstream application hints at fully automated low-level motion planning and dexterous manipulation, enabling realization of household robots.

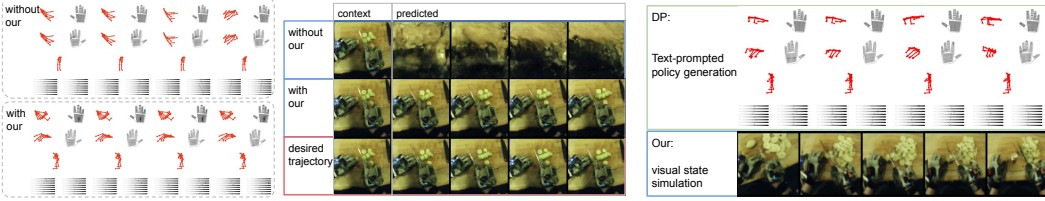

Figure 10: **Left:** Results on goal-conditioned policy optimization. **Right:** Results on long-term task planning.

## 5 CONCLUSION

In this work, we introduce the concept of multisensory interaction to fine-grained generative simulation. We focus on the the task of learning an effective multisensory feature representation to effectively control a downstream video generative simulator. Our proposed multimodal feature extraction paradigm along with our regularization scheme produces action feature vectors capable of accurately controlling the generative simulator and robust to missing modalities at test time. We

conduct extensive comparisons, ablation experiments, and downstream applications to showcase the merits of our method. We hope our work brings insights and inspirations to the research community.

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

# 6 APPENDIX

**Disclaimer.** This is a research work where the primary focus is introducing a new task and a method to learn effective multimodal representation for generative simulation. The goal of this work is **not** to provide production-level video resolution. We devise our multimodal feature extraction as generic to be combined when stronger video generation backbone is invented. We hope our work can inspire future research works and industrial efforts to build foundational digital twin of our world with fine-grained control. We hope our work can be used to scale with abundant resources.

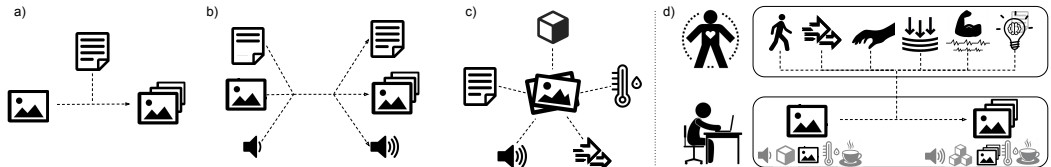

Figure 11: Existing multimodal learning tasks focus on vision-language binding, cross-modal retrieval, and modalitiy anchoring focuses on mining the similarity between different modalities of data (a, b, c) (Yang et al., 2023; Ruan et al., 2023; Girdhar et al., 2023). On the other hand, the task of multisensory action conditioned generative simulation (d) need to understand the unique aspect of each interoceptive action modalities (top) and combine the synchronously to change the exteroception of the external world (bottom).

## 6.1 NOTATION AND ADDITIONAL PIPELINE FIGURE

We summarize the notation used in our paper in Table. 2, and we provid additional pipeline Fig. 12.

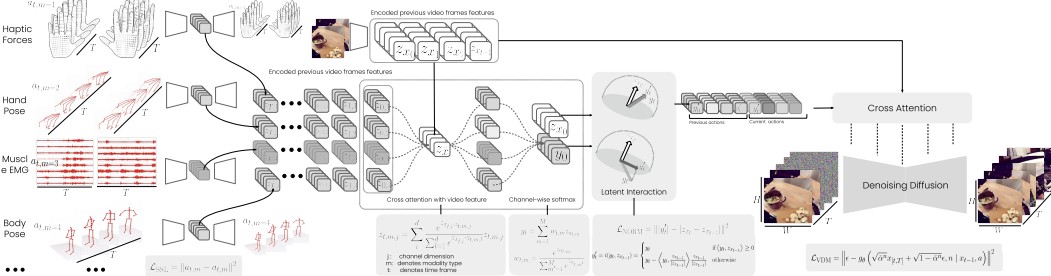

Figure 12: Additional pipeline figure.

| | |
|---|---|
| time frame | $t$ |
| history horizon | $[0, t-1]$ |
| future frames | $[t-1, T]$ |
| video frame | $x_t$ |
| encoded video frame | $z_{x_t}$ |
| action modality | $m$ |
| action modality signal | $a_{t,m}$ |
| encoded action modality $m$ signal at time step $t$ | $z_{t,m}$ |
| j-th dimension of encoded action modality $m$ signal at time step $t$ | $z_{t,m,j}$ |
| cross-modal feature | $y_t$ |
| regularized cross-modal feature | $y'_t$ |

Table 2: Notation Chart

## 6.2 MODEL SIZE

We report the modules of our model in Table. 4. We can see that the multimodal action signal module is fairly small compared to the video module. Each signal average to around 18044828 parameters which is only 5 percent of the total model weights. The lightweight action signal heads highlights the advantage of our method for low computational cost added for each action signal modality

| module | parameter count | percentage of total |
|---|---|---|
| signal expert encoder | 43780932 | 0.13 |
| signal projection | 11537408 | 0.03 |
| signal decoder | 28398382 | 0.08 |
| signal Total | 83716722 | 0.25 |
| video model | 252380168 | 0.75 |
| total model | 336096890 | 1.00 |

Table 3: Parameter Count on $64 \times 64$ model.

| module | parameter count | float16 in MB | float32 in MB |
|---|---|---|---|
| policy network (to be deployed on edge devices) | 120690484 | 241MB | 482 MB |

Table 4: Parameter Count for the policy network model used in Downstream application section.

## 6.3 CROSS SUBJECT TESTING

We report the cross subject testing on three different subjects in the ActionSense dataset, result can be found in Table. 5.

Table 5: Cross Subject Testing

| Method | MSE $\downarrow$ | PSNR $\uparrow$ | LPIPS $\downarrow$ | FVD $\downarrow$ |
|---|---|---|---|---|
| subject 2 | 0.115 | 15.8 | 0.301 | 206.7 |
| subject 4 | 0.112 | 16.0 | 0.282 | 204.6 |
| subject 5 | 0.110 | 16.0 | 0.276 | 203.5 |

## 6.4 RELATED WORK

**Learning Multi-Modal Representations.** Learning shared representations across various modalities has been instrumental in a variety of research areas. Early research by De Sa et al. de Sa (1994) pioneered the exploration of correlations between vision and audio. Since then, many deep learning techniques have been proposed to learn shared multi-modal representations, including

vision-language Joulin et al. (2016); Desai & Johnson (2021); Radford et al. (2021); Mahajan et al. (2018), audio-text Agostinelli et al. (2023), vision-audio Ngiam et al. (2011); Owens et al. (2016); Arandjelovic & Zisserman (2017); Narasimhan et al. (2022); Hu et al. (2022), vision-touch Yang et al. (2022); Li et al. (2023b), and sound with Inertial Measurement Unit (IMU) Chen et al. (2023). Recently, ImageBind Girdhar et al. (2023) and LanguageBind Zhu et al. (2023) demonstrate that images and text could successfully bind multiple modalities, including audio, depth, thermal, and IMU, into a shared representation. However, these previous efforts take bind-all fuse-all perspective, which takes away many of the inherent differences brought by various sensory modalities. Our work takes a different perspective. By differentiating between the active and passive senses, we allow a bilateral model to arise and capture the interaction between the two. The prior fuse-all strategy also overshadows an inherent need in multi-modal representation learning, which is interaction. We propose a representation learning scheme to capture the nature of multi-modal interactions.

**Learning World Models.** Learning accurate dynamics models to predict environmental changes from control inputs has long challenged system identification Ljung & Glad (1994), model-based reinforcement learning Sutton (1991), and optimal control Åström & Wittenmark (1973); Bertsekas (1995). Most approaches learn separate lower-dimensional state space models per system instead of directly modeling the high-dimensional pixel space Ferns et al. (2004); Achille & Soatto (2018); Lesort et al. (2018); Castro (2020). While simplifying modeling, this limits cross-system knowledge sharing. Recent large transformer architectures enable learning image-based world models, but mostly in visually simplistic, data-abundant simulated games/environments Hafner et al. (2020); Chen et al. (2022); Seo et al. (2022); Micheli et al. (2022); Wu et al. (2022); Hafner et al. (2023). Prior generative video modeling works leverage text prompts Yu et al. (2023); Zhou et al. (2022), driving motions Siarohin et al. (2019); Wang et al. (2022), 3D geometries Weng et al. (2019); Xue et al. (2018), physical simulations Chuang et al. (2005), frequency data Li et al. (2023c), and user annotations Hao et al. (2018) to introduce video movements. Recently, Yang et al. Yang et al. (2023) proposes Unisim, which uses text conditioned video diffusion model as an interactive visual world simulator. However, these prior works focus on using text as condition to control video generation, which limits their ability to precisely control the generated video output, as many fine-grained interactions and subtle variations in control are difficult to be accurately described only using text. We propose to use complementary multi-sensory data to achieve more fine-grained temporal control over video generation through multi-sensory action conditioning.

## 6.5 IMPLEMENTATION DETAILS

**Network Architecture Detail** We use the open-source I2VGen (Zhang et al., 2023) video diffusion network as our backbone. We modify original I2VGen to take pixel space data by changing the input channel to 3 (originally set to 4) and change input image size to $64 \times 64$. We keep all other parameters unmodified, and vary the input condition type. We note that single condition models that only use image **or** text such as Stable Diffusion (Rombach et al., 2021) and etc. are not sufficient for our purpose.

All text input are encoded using CLIP text encoder from the open-source OpenClip (ope) libary. Images are encoded also using OpenClip Image encoder. Specifically, we use the *ViT-H-14* version with *laion2b_s32b_b79k* weights. Please refer to the original papers (Zhang et al., 2023; ope) their architecture details. We describe the architecture of the remaining modules of our model.

Signal specific encoder heads for hand pose, body pose, emg uses the same MLP architecture with different input dimension. The input dimension for hand pose is $24 \times 3 \times 8$, body pose is $28 \times 3 \times 8$, emg is $8\times$, hand force is $32 \times 32 \times 8$. MLP is composed of four layers, with GeLU activation. We set the hidden and output dimension of 128. We apply a dropout with p=0.1, with batchnorm applied in the first two layers. All encoded signals then goes through a three-layer MLP projection head to project the encoded feature to the same space $\mathbb{R}^{1024}$ as the clip image feature. The projection MLP also uses GeLU activation with dimensions of [input_dim, 512, 768, 1024]. We apply batchnorm after the first layer. The set of features are then aggregated across the sensory modalities and masked by a softmax in the modality dimension.

For the latent interaction layers, we use each context frame vector and the action vector for the corresponding timestep $t$ for the context frame feature regularization, we use the aggregated average context frame feature $z_{x_{\bar{t}}}$ to form the context vector for the current action features.

For the experiments comparing to unimodal action sensories, we use our own method for encoding these modalities and conditioning video model. For the sensory modalities of muscle EMG and hand forces, there lacks research works concerning the senses of muscle activation and haptic forces. For hand poses, most works concerning hand poses tackle the task of detection of hand regions from videos (Qu et al., 2023; Zhang et al., 2022; Kwon et al., 2021). Therefore they also cannot be directly adapted to compare with our work. For this reason, we use our own method for encoding these modalities and conditioning video model.

For experiments on down stream application, we follow the original diffusion policy implementation. The image prompted DP (Sec. 4) uses ResNet (He et al., 2016)-18 image encoder, and the text prompted DP (Sec. 4) uses OpenClip (ope) text-encoder. We modify the original 1D UNet to be four layers with hidden dimensions set to [128, 256, 512, 1024]. The dimension of action space comes to 2292, with two hand poses $24 \times 3 \times 2$, one body pose $28 \times 3 \times 1$, two arm muscle emg $8 \times 2$, two hand forces is $32 \times 32 \times 2$.

**Hardware, Software, Training Setup** We use a server with 8 NVIDIA H100 GPU, 127 core CPU, and 1T RAM to train our models for 15 days. We implement all models using the Pytorch (Paszke et al., 2019) library of version 2.2.1 with CUDA 12.1, and accelerator (Gugger et al., 2022) and EMA (Karras et al., 2023b) . We train our models with batch size of 18 per GPU. We use the Adam (Kingma & Ba, 2015) optimizer with learning rate of $1e - 4$ and betas $(0.9, 0.99)$, ema decay at 0.995 every 10 iterations.

**Experimental Setup** The ActionSense (DelPreto et al., 2022) dataset does not contain the detailed text description used in Sec. 3.1. We generate these text descriptions by using several metrics. We augment the original dataset by resampling video frames, three-ways, every frame, every other frame, and every three frames. We add description of `slow in speed` to the first chunk of data, and `fast in speed` to the third chuck of data. Additionally we also calculate the average hand force magnitude for every task. If the hand force sequence contains frames that are significantly larger than the average frame we add `holding tightly` and add `holding gently` to the lowest force data sequences.

### 6.6 Discussion of Limitations and Future Work

Our experiments are conducted on datasets of human actuation and activities. Ideally, it would be interesting to see the deployment of planned and optimized policies on real humanoid robots with similar multi-sensory capabilities. Because we currently do not have such hardware setup that enables dense force readings on human-hand-like robotic hands or various other fine-grained interoceptive modalities on humanoid robots. We leave this direction for a future research.

There are other passive exteroceptive senses that can be combined with vision, such as depth, 3D and audio etc. One can directly leverage a multi-branch visual-audio or visual-depth UNet diffusion model as the backbone to achieve such multi-modal experoception responses. However, due to limited availability of such data, we leave this direction as future work.

Additionally, because of limited computational resources, we limit our video diffusion model to be very low resolution. However, one can employ upsampling approaches to map low-resolution video predictions to higher resolution. Our work is less concerned with the specifics of image quality but more with the application of using multi-sensory interoception data. Therefore, we leave the study of low-cost video upsampling or better video diffusion backbone as future work.

### 6.7 Additional Experiments and Discussion

#### 6.7.1 Text as addition to multisensory actions

We are also interested in learning whether multi-sensory action can entirely replace text as condition. We integrate an additional text-encoder head to the MoE feature encoding branches to incorporate simple text phrases, *e.g.* `cut potato`. The encoded text features are aggregated with other multi-sensory action features in the same manner as described in Sec. 2.1. We use the pretrained OpenClip (Ilharco et al., 2021) text encoder to encode text in all baselines and our model.

As depicted in the bottom half of Figure. 8, when multiple objects (pan and plate) appear in context image and when the action trajectory can be applied to both objects, the network is uncertain about

which object to apply the action. It cleans the plate instead of the pan. When we add text description `clean pan` as an extra piece of information, ambiguity is removed and accurate video can be generated. We also observe that when the context frame is not ambiguous, multi-sensory action provides enough information to generate accurate video trajectories. Adding additional text feature induces a temporal smoothing effect generating similar images across frames.

### 6.7.2 ADDITIONAL RESULTS ON TEST-TIME ROBUSTNESS

Table 6: Testing with single modality available

| Method | MSE ↓ | PSNR ↑ | LPIPS ↓ | FVD ↓ |
|---|---|---|---|---|
| Hand pose | 0.121 | 14.6 | 0.309 | 210.2 |
| Hand force | 0.117 | 14.7 | 0.307 | 208.0 |
| Body pose | 0.123 | 14.6 | 0.310 | 210.5 |
| Muscle EMG | 0.132 | 13.9 | 0.312 | 214.8 |
| All sensory used | 0.110 | 16.0 | 0.276 | 203.5 |

As we see from the Table. 6 that when one modality is provided, our model can still produce higher prediction accuracy compared to text-based models or single-model models. Comparing this result with Table. 3a shows that our proposed multsensory action trainiing strategy induces higher quality action feature compared to training with a single modality. This comparison indicates that through implicit association between different modalities, both feature alignment and information presevation is achieved. That is, the complementary information is preserved in the feature representation such that when only one action modality is provided, the model might have access to commonly co-activated feature dimensions and thus produce better result than training with single modality.

To provide a comprehesive set of ablation studies on testing with missing modalities, we show Table 7 that includes all possible pairs of modalities used during testing. The results in Table. 7 along with Table. 6 and Table. 1b makes a comprehensive study cross all possible ablated experiments. We can from Table.7, that the model achieves better performance when different aspect of information is provided .

Table 7: Testing with paired modality available

| Method | MSE ↓ | PSNR ↑ | LPIPS ↓ | FVD ↓ |
|---|---|---|---|---|
| Hand Pose and Hand Force | 0.115 | 14.9 | 0.304 | 206.4 |
| Body Pose and Muscle EMG | 0.122 | 14.6 | 0.309 | 210.1 |
| Hand Force and Muscle EMG | 0.117 | 14.7 | 0.307 | 207.6 |
| Hand Pose and Body Pose | 0.113 | 15.0 | 0.297 | 206.2 |
| All sensory used | 0.110 | 16.0 | 0.276 | 203.5 |

### 6.7.3 EXAMPLES OF FINE-GRAINED CONTROL

We can see from Fig. 13 where hand force together with hand pose helps accurately controls the timing of the hand grabbing the pan.

### 6.7.4 ADDITIOANL QUALITATIVE RESULTS ON OTHER DATASET

To show that our proposed method is generic is not designed for the ActionSense DelPreto et al. (2022) dataset, we conducted an experiment by directly applying our proposed approach on another dataset, H2O dataset (Kwon et al., 2021). H2O dataset (Kwon et al., 2021) is a unimodal action-video dataset that includes paired video and hand pose sequences. We show experiment on H2O (Kwon et al., 2021) to demonstrates that our system is generic, not dataset specific, and can achieve reasonable performance when operating on other datasets. Qualitative results are provided in Fig. 14. These results indicate that our model is capable of training and testing on unimodal action datasets, highlighting its generalizability beyond the ActionSense DelPreto et al. (2022) dataset. This

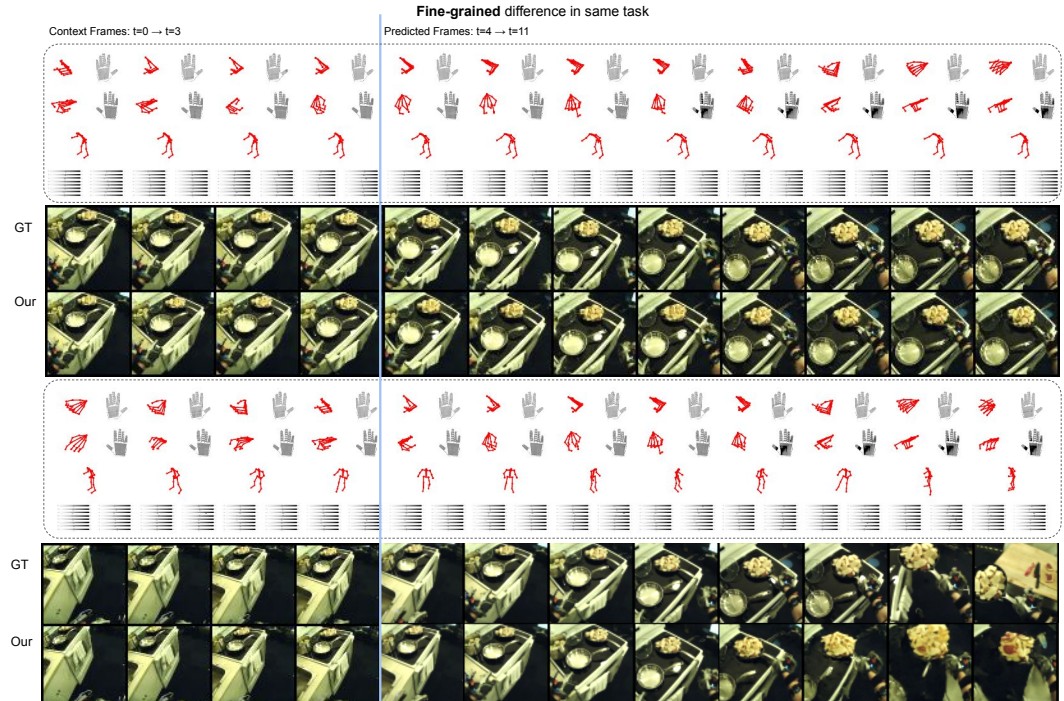

Figure 13: Temporally fine-grained control

demonstrates that our method is not specifically tailored to ActionSense DelPreto et al. (2022) and can adapt to various scenarios.

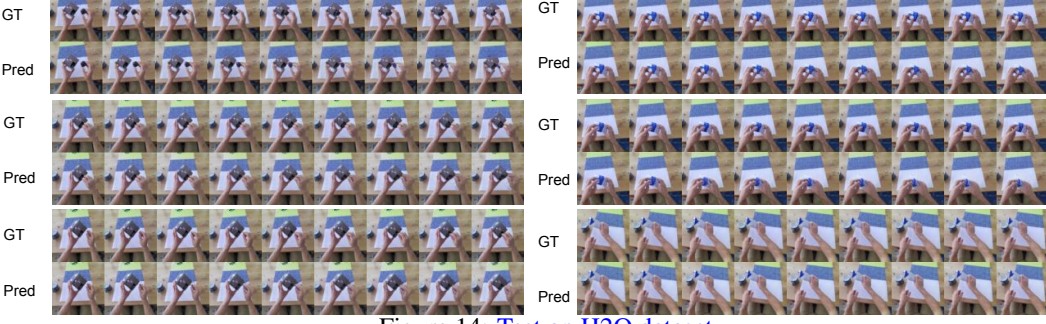

Figure 14: Test on H2O dataset

## 6.8 ADDITIONAL QUALITATIVE RESULTS

Additional Qualitative Results are shown in Fig. 15, Fig. 16, and Fig. 17. Fig. 15 and Fig. 16 show additional qualitative results of context frames and predicted video frames from our proposed multisensory action signals. Fig. 17 shows demonstrations of failure cases, policy optimization, and long-trajectory planning. We show one most recent context frame and the eight prediction frames. Fig. 17 shows results paired in two rows, where he top row shows ground truth trajectory the bottom row shows predicted trajectory. We show the failure cases on the top right section. Common failure cases include false hallucination of environment with large motion. Failure to identify object with similar apperance to background. The wooden chopboard gradually disppear into the wooden table background and fails to pick it up in simulation. Failure in identify object to act on (also hallucates pan handle on plate and cleaning the plate). The last five rows in Fig. 15 show additional results on down stream tasks of policy planning, shown in the middle rows, and long-trajectory simulation, show in the bottom row.

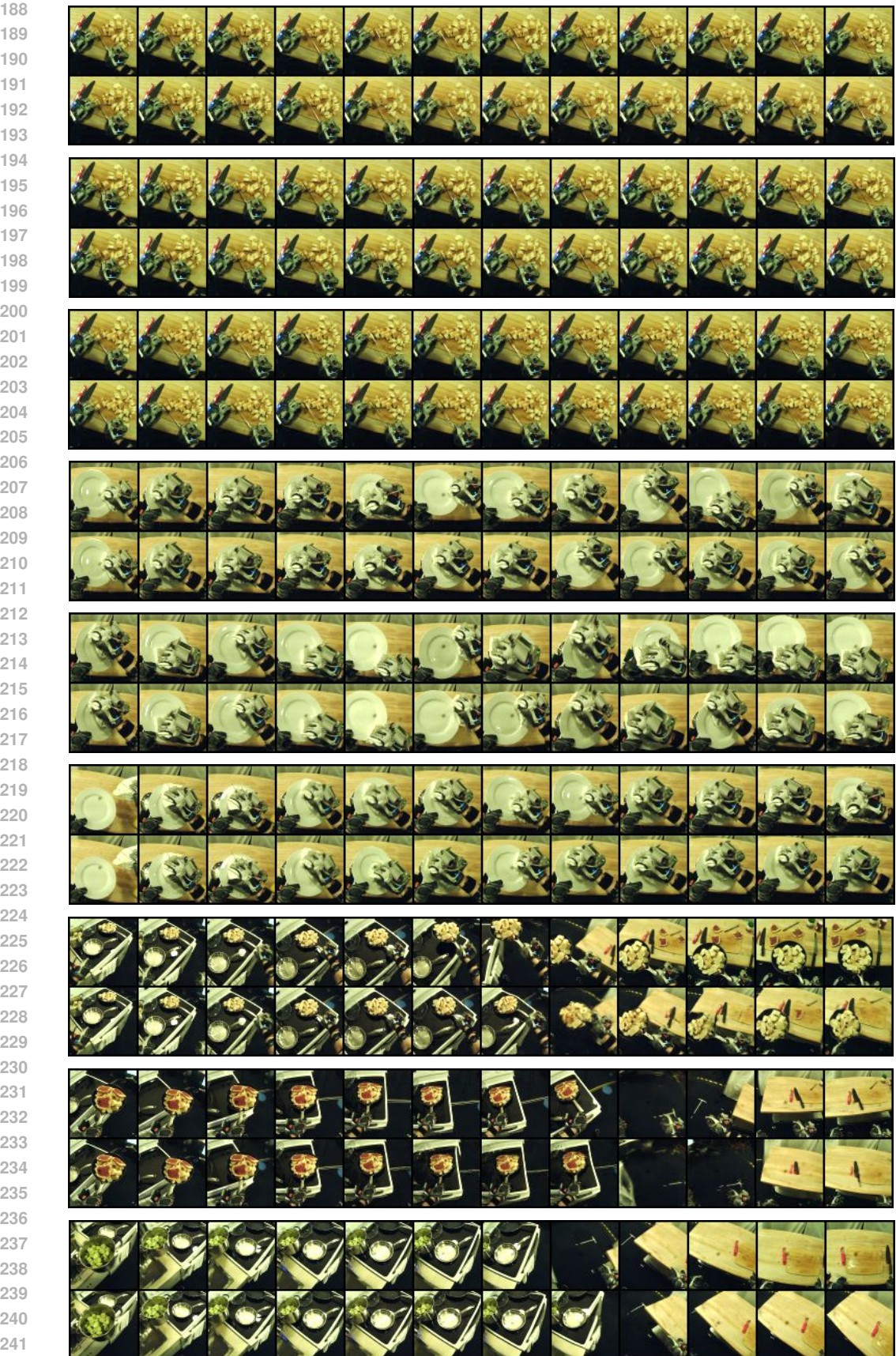

Figure 15: Additional qualitative results

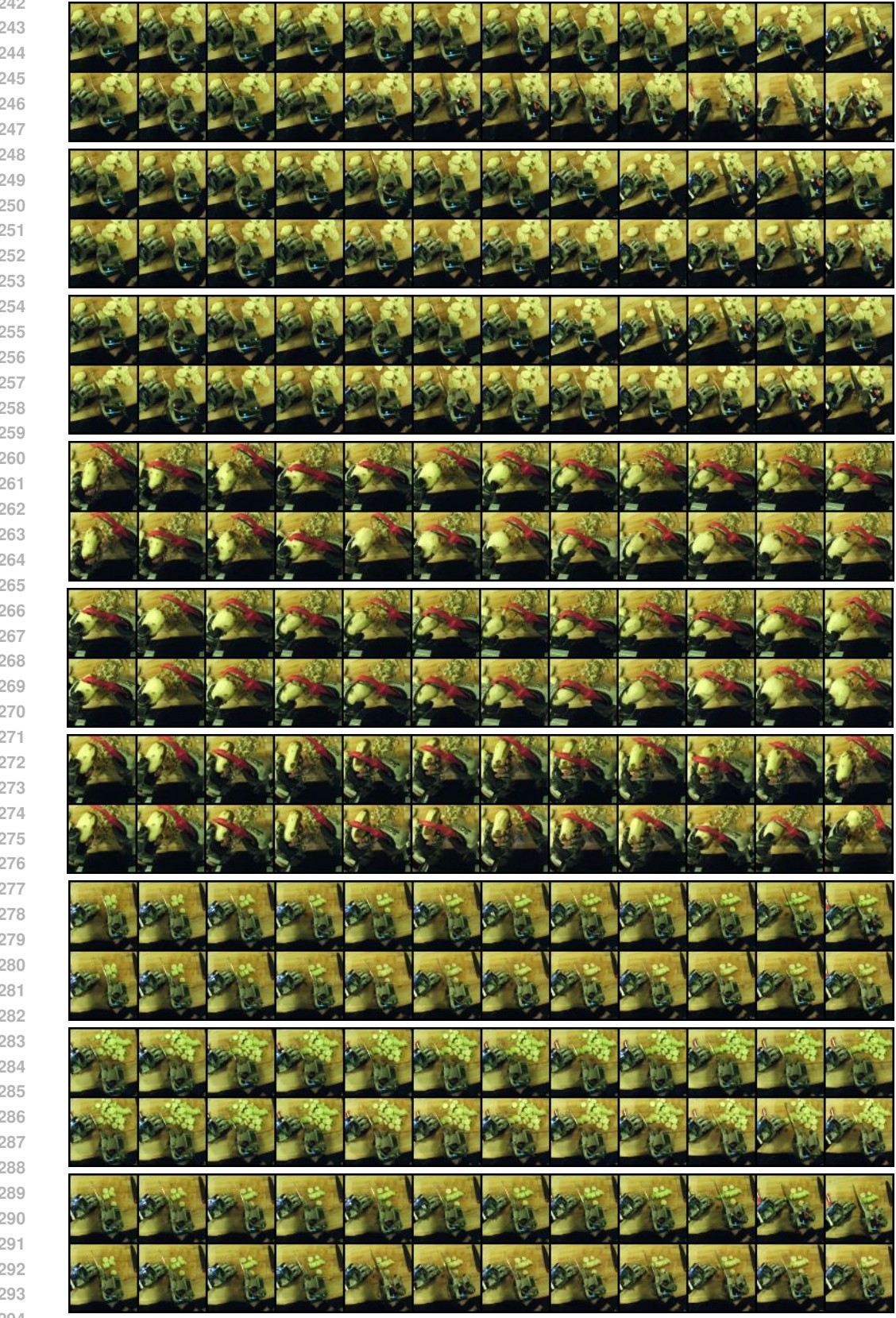

Figure 16: Additional qualitative results

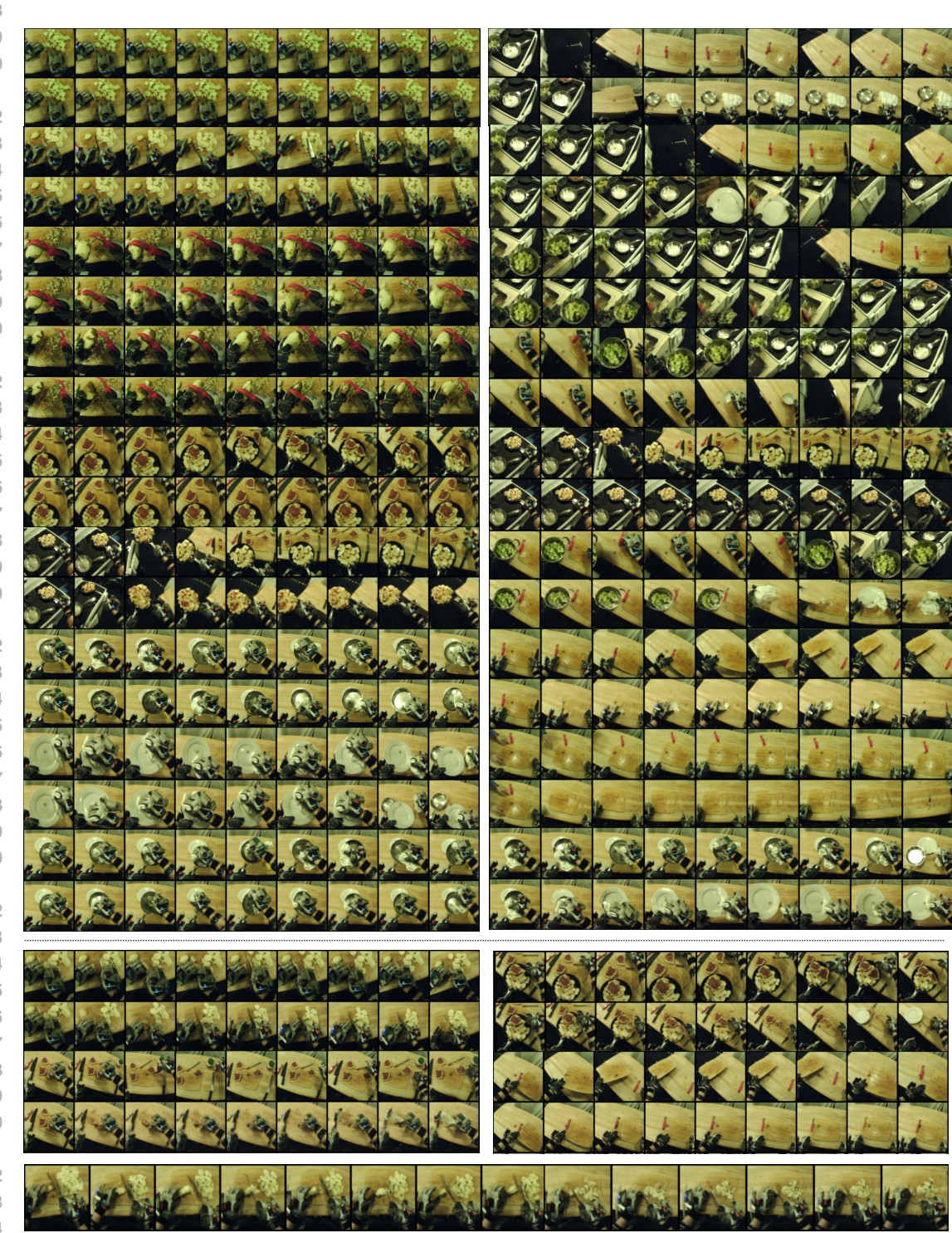

Figure 17: **Top left:** Additional qualitative results. **Top right:** Failuare cases. **Middle left and right:** Additional results on policy optimization. **Bottom:** long-trajectory policy planning.

