# OpenReview forum: "Skin, Muscles, and Bones in MultiSensory Simulation"
_ICLR.cc/2025/Conference — Submitted to ICLR 2025_

### Official Review · Reviewer_qMjQ · 2024-10-29

**Soundness:** 3
**Presentation:** 3
**Contribution:** 3
**Rating:** 8
**Confidence:** 3

**Summary:**

The authors proposed a multisensory action presentation learning for generatively simulating videos with fine-grained manipulation. They introduced a multimodal feature extraction paradigm to align modalities including haptic forces, muscle stimulation, hand and body poses. They also proposed a generative simulation method by using the aligned multimodal action representation for fine-grained control of a video diffusion simulator. They evaluated the propose method with extensive ablation studies and comparison with previous work, and also demonstrated the applications in policy learning and planning.

**Strengths:**

This work is well-motivated to leverage multimodal sensory measurements to simulate fine-grained control. The problem is well-formulated and the proposed approach is well-explained with great details. The work has thorough and concrete experiments and evaluations.  Both  quantitive and qualitative results are presented with analysis. I believe the system can be adapted by further researchers.

**Weaknesses:**

I assume all tests were conducted within the same domain as the training set, such as the data were recorded when the demonstrators were doing the similar tasks with similar tools, or equipped the sensors in the same way.  I wonder how generalizable the proposed method can be? Such as what if having a different background, using a tool with different shape or color, cutting different vegetables/fruits?

**Questions:**

1. Section 2.1. About the cross attention: What is z_{t, m, j} and is it used anywhere else?
2. Section 2.2 Relaxed Hyperplane Interaction: The sentence “A geometric interpretation of the latent interaction… depicts two space partitioned by a hyperplane defined by …” was confusing to me. Could you explain it with more details?
3. Section 2.2 Relaxed Hyperplane Interaction: L_{VDM} was used before its definition.
4. Fig 3 and Fig 6. (a) Please highlight the best results for each column.  (b) Please explain what LPIPS is in the figure caption. Note: it would be great if all figures are self-contained with short explanations in caption.
5. Please highlight the best/the best two results for each column in Table 1.
6. Section 3.1 Comparison with Text-conditioned Simulation: typo “Phrasse” -> “Phrase”
7. How many tasks are there in the dataset? And what are these?
8. Figure 15: Should Top right be Failure cases and Top left be additional qualitative results?

---

> ### Author Response · Authors · 2024-11-21
> **Author Response**
>
> We would like to thank the reviewer for the effort in reviewing our work and for the constructive feedback. We have updated our draft with 1) clarified notation  2) change in paragraph structure ensuring all terms are defined 3) updated figure caption 4) highlighted best performance 5) corrected typos
>
> **OOD data** This is an interesting question. We conduct one experiment to show how our work can handle specific out-of-distribution (OOD) scenarios, specifically through fine-tuning. As we lack the sensor hardware to collect similar data. For this experiment, we modified the original ActionSense dataset to create OOD data.  We first use segmentation foundation modal, LangSAM [1], to extract segmentation masks for "potatoes, " and then we recolored the "potatoes" to appear as "tomatoes." Since the video model had not encountered red vegetables or fruits during training, we fine-tuned our pretrained model on a small dataset of approximately 600 frames (30 seconds) and evaluated it on the test split of this "tomato" data. The data creation procedure and results on this experiment can also be found on our [supplementary website](https://sites.google.com/view/iclrsubmissionmultisensorysim) . The results show that the model achieves reasonable performance after fine-tuning. While we acknowledge that robust in-the-wild generalization requires training on larger-scale datasets with diverse domain coverage, this experiment illustrates a practical use case for addressing OOD data. Specifically, it demonstrates that by collecting a small, specialized dataset, our pretrained model can be effectively fine-tuned to adapt to new domains.
>
> **Hyerplane** Thank you for this question. In the latent projection interaction paragraph of Sec 2.2, we define the interaction feature to be the orthogonal component of action feature from the context feature, $y'\_t = y\_t - \left< y\_t,\frac{z\_{x\_{t-1}}}{|z\_{x\_{t-1}}|} \right> \frac{z\_{x\_{t-1}}}{|z\_{x_{t-1}}|}$. From a pure linear algebra perspective, this equation is precisely the vector-plane projection equation. It is easier to imagine the [vector relationships in $\mathbb{R}^{3}$](https://www.maplesoft.com/support/help/Maple/view.aspx?path=MathApps/ProjectionOfVectorOntoPlane), where the direction of the plane is defined by a normal vector $z\_{x\_{t-1}}$. Any plane partitions the entire space $\mathbb{R}^3$ in to two spaces, one that lies on top of the plane and one on the bottom of the plane. In higher dimensions $\mathbb{R}^{d}$, this plane becomes a hyperplane that also partitions the space of $\mathbb{R}^{d}$ into two spaces.  We use these geometric relationships between vectors and normal directions to define a relaxed version of interaction vector y' where we treat the positive space (on top of the hyperplane) to be where action takes affect on the context and clip/project the interaction when it resides below the context hyperplane.  Essentially, hyperplane here is the context feature vector treated as normal direction vector.
>
> **Tasks in ActionSense**
> The original ActionSense Dataset includes peeling cucumber / potato, slicing bread / banana / potato / cucumber, cleaning pan / plates, and fetching and transferring tableware, such as pan / pots. We do find that some of the tasks enlisted on the ActionSense paper, such as pouring water and spreading jam on bread etc., are not collected across subjects and only very small amount of data is collected on one subject.
>
> We thank you once again for your encouraging feedback, and we hope that you will find all your suggestions well addressed in the revision.
>
> [1] https://github.com/luca-medeiros/lang-segment-anything

---

> ### Comment · Area_Chair_omip · 2024-11-25
>
> Dear Reviewer,
>
> Please provide feedback to the authors before the end of the discussion period, and in case of additional concerns, give them a chance to respond.
>
> Timeline: As a reminder, the review timeline is as follows:
>
> November 26: Last day for reviewers to ask questions to authors.
>
> November 27: Last day for authors to respond to reviewers.

---

### Official Review · Reviewer_Xsn3 · 2024-11-01

**Soundness:** 3
**Presentation:** 3
**Contribution:** 3
**Rating:** 6
**Confidence:** 3

**Summary:**

This paper introduces a multisensory approach for generating videos, where proprioceptive, kinesthetic, force haptic, and muscle activation data are used to condition a video diffusion model that synthesizes realistic frames. Key contributions include a multimodal feature extraction paradigm and a novel regularization scheme to enhance the causality and alignment of action trajectories in the latent space. Using the ActionSense dataset, the authors demonstrate that multisensory input enhances prediction accuracy and temporal consistency compared to unimodal conditioning.

**Strengths:**

- This approach addresses significant gaps in current video prediction methods, particularly for videos involving fine-grained motor movements.
- The regularization applied in the latent space is shown to improve model performance, potentially enhancing prediction stability and the temporal alignment of generated frames.
- This framework could advance simulators for fields requiring multisensory input, such as robotics or motor skill training. Integration of proprioceptive and haptic data improves video generation performance, which could support downstream tasks such as adaptive control policy development.

**Weaknesses:**

- The explanation of core processes—multisensory feature fusion, latent regularization, and video diffusion configuration—is often vague. Terms and subscripts are not well defined, making it difficult to follow some derivations and mechanisms. Similarly, the introduction and other parts of the text can be improved. Additionally, figures lack arrows or labels that would clarify data flow, and some of the text within illustrations is too small.
- It's not clear the effectiveness of multisensory integration compared to the other proposed mechanisms. In particular, the ablation study reveals a minimal performance drop when some sensory modalities are removed at test time, suggesting unexpected redundancy. Table 2 shows that the model with a single modality (e.g., hand-force only) can still achieve high performance, raising questions about the true necessity of multisensory inputs compared to other mechanisms at play. For example, hand and body pose data seem to be crucial but can be removed at test-time with minimal impact on LPIPS scores, which is counterintuitive and requires further investigation.
- This approach depends heavily on multisensory datasets, which are often costly and difficult to obtain in real-world settings. Applications would require large-scale "in-the-wild" data to achieve generalization, limiting the method’s accessibility.
- The paper primarily compares performance variations within its own framework but lacks comparisons to integrate multi-sensory modality with other baselines (e.g., Mutex, LanguageBind, ImageBind).
- The low-level policy optimization with the current simulator is not clearly articulated, specifically the difference between the baseline and the policy with the simulator.
- While the multisensory conditioning approach improves the performance, the model would benefit significantly from the ability to predict multisensory outputs in addition to visual frames. Predicting proprioceptive, kinesthetic, and haptic feedback could open up for multiple applications, where accurate multisensory feedback is essential.

**Questions:**

- Consider revisiting the title to more accurately reflect the multisensory conditioned video generation aspect, as the current title may be misleading.
- Why were the specific baseline models chosen, and how do they differ fundamentally from the current method? If these methods were augmented with multisensory data, would they perform comparably?
- In unimodal tests, is cross-attention with video features and action regularization also employed? Is the number of modalities the only thing that differs in these comparisons?
- How does performance change when multiple modalities are removed (similar to Table 2)? In particular, when does the model leverage co-interaction between modalities? It would be good to have a more comprehensive ablation study showing performance across possible combinations of modalities.
- How does computational efficiency scale with additional sensory inputs? How does that affect computation time and memory usage?
In policy optimization, how does the policy generate frames when the proposed simulator is removed? It’s not clear the comparison with the baseline policy
- Did the authors consider adding the capability to predict multimodal sensory outputs? What are the potential challenges? This would significantly extend the application and contribution of this work.

---

> ### Author Response · Authors · 2024-11-20
> **Author Response (1/4)**
>
> We thank the reviewer for the feedback. We have revised our draft with:
> 1) notation clarification, notation lookup table.
> 2) added arrows to method figure, additional supplementary method figure,
> 3) additional experimental result analysis discussion comparing to the baseline methods.
> 4) additional discussion comparing training and testing with missing modalities, highlighting the advantage of training with multisensory modalities,
> 5) comprehensive ablation stufies across all possible combinations of modalities,
> 6) computational efficiency model size table, showing that the proposed multimodal representation method is lightweight,
> 7) clarification ob low level policy optimization setup.
>
> **Manuscript revision for clarification**
>
> - **Revise title** Thank you for this suggestion, we are considering revising the title to one of the following, we would love to hear your and all reviewers' input on the revision choice of the paper title
>
>   - Multisensory Representation Learning for MultiModal Video Simulation, which emphasizes the multisensory representation learning prespective.
>   - Skin Muscles and bones in Multisensory Video Simulation, which emphasizes the sensory modalities used.
>
> - **Pipeline figure, term definition**
> We have revised the problem statement section to enhance clarity by refining the terms, notation, and subscripts used throughout the paper. Additionally, we have included a **notation lookup chart**, provided an **additional pipeline figure**, Sec A6.1 and Fig.12 in Appendix. We updated the original figure incorporating arrows to better clarify our approach. We hope these updates address any ambiguities and make our method more accessible to readers.
>
> - **Clarity of text** Detailed implementation detail including multisensory feature fusion (line 927-932), latent regularization (line 934-936), and video diffusion configuration (line 944 - 949) are already included in the Implementation Details section in the appendix in the original draft.
>
>     For the comment: **the introduction and other parts of the text can be improved,** we would like to kindly ask the reviewer to refer to us the specific parts , i.e. paragraph or line number, so we can better address the concern accordingly and improve our manuscript.
>
>
> **Baseline Methods**
>
> - **How are the baseline method chosen?** These baseline models are commonly used multisensory learning schemes. They are designed for aligning multimodal data and extracting multimodal representation.
>
> - **Would these methods perform better when augmented with multisensory data?** The comparison conducted are already using the same multisensory data, as the baseline methods are multimodal representation learning schemes. In the comparisons with the baseline, the only difference is the how multisensory representation is extracted from the data, input data, i.e. tactile, muscle EMG, body pose, hand pose, context frame, and video backbone are the same.
>
> - **lacks comparisons to integrate multi-sensory modality with other baselines (e.g., Mutex, LanguageBind, ImageBind).**
> The entire section 3.2 is dedicated to comparison with these baselines, results can be found in Figure 5, Figure, 6, Table 6. from line 324-377 on page 7 and 8.
>
> - **How are baseline methods different from the proposed method?**
>   - The discussion regarding these differences can be found in our manuscript: in the third paragraph of the introduction (lines 36–43), the remark in the method section (lines 140–144), and the experimental results discussion section (lines 359–398). Additional comparisons with existing methods are provided in the related work section (lines 847–861). Furthermore, we have updated our manuscript to include an expanded discussion analyzing qualitative comparisons between our approach and baseline methods in Section 3.2 (lines 389–406).
>   - In short, the key difference between the existing approaches and our approaches lies in how multimodal features are aligned and represented. The baseline approaches often employ contrastive loss (LanguageBind, ImageBind) or directly regressing cross modal features using MSE loss (Mutex). Contrastive losses magnifies the similarity between features, exploring the substitutional property of crossmodal features, which is well suited cross-modal retrieval tasks, but not the task we propose: multimodal action signals for video simulation. This task necessitates not only aligning features through shared substitutional information but also preserving the unique complementary information inherent in each modality. Therefore, we propose an implicit alignment approach that uses channel-wise attention between various multisensory action signals and encoded context features and channel-wise softmax for cross-modal aggregation. Detailed explanations of our method can be found in the aforementioned sections of our paper.
>
>      We hope this clarifies how our approach distinguishes itself from existing methods and addresses your concerns.

---

> ### Author Response · Authors · 2024-11-20
> **Author Response (2/4)**
>
> **Minimal performance drop of testing with missing modalities** Thank you for this question, we would like to clarify the following.
>   - **good performance removing modalities during test-time means multimodal is not needed?**  We respectfully disagree with this perspective. This experiment shows that our model trained with all modalities is robust to missing modalities during test-time. We would highlight a key difference between two experiments, training with ablated modalities Table 1 (a) and testing with missing modalities Table 1 (b). In the latter experiment, the model is trained with all modalities, enabling it to leverage correlations between modalities to compensate for missing inputs at test time. Conversely, in the former, the model is trained with only a subset of modalities, leading to less accurate predictions. The performance gap between Table 1(b) and Table 1 (a) precisely shows the advantage and necessity of training with multimodal action signals. Related discussion can be found in the introduction (line 46- 48) and remark paragraph (line 140 - 145) in the method section. We would like to point out that when majority of such information is removed together as shown in Table 6 ( originally 1) and Table 7 and in the Appendix section, the model suffers from more significant performance drop.
>
>   - **Reason for better performance obtained with multimodal training:** When the model is trained with all modalities present, the model can access more complete action information for learning different modes of action, and thus induce stronger performance compared to trained with limited modalities. Because our proposed channel-wise attention and channel-wise softmax integration help to correlate the these substitutional latent channels across the different modality of signals, and thus allowing them to compensate for the missing modalities during test time.
>
>   - **Reason for good performance testing with missing modalities:** To aid the understanding of robustness to missing modalities during test-time, we make an inaccurate analogy to PointNet[1]. There are many points in the point cloud, but only few points are critical to decide the shape category. PointNet is known for robustness, the max operator in PointNet selects these critical information for the entire point cloud. Similarly in our work, the cross-channel softmax selects most representative information for manipulation, *obscuring the source modality* that it came from, which is the main reason for the robustness towards missing modality.
>
>   - **Redundancy in Representation** A more philosophical aspect to representation learning is that redundancy is not necessarily a bad thing, on the contrary, it helps with generalization [2], which is consistent with the findings of our work. Many networks have redundancy in representation, kernels in CNNs, attention in Transformers, and etc. We would like to ask the reviewer to further clarify in which way our ability to robustly handle missing modality during test time / potential redundancy in the representation is bad/weakness (also given that our work is a simulator does not need to run / store on edge robot devices).
>
> **Comprehensive ablation across all possible combinations of modalities.**  We have added additional experimental results testing on all possible combination of modalities, The additional results are included in Table 7 in the revised draft. Table. 7 along with
> Table. 6 and Table.1b makes a comprehensive study cross all possible ablated experiments.
>
> | Testtime modalities | MSE | PSNR | LPIPS | FVD |
> | --- | --- | --- | --- | --- |
> | Hand Pose & Hand Force | 0.115 | 14.9 | 0.304 | 206.4 |
> | Body Pose & Muscle EMG | 0.122 | 14.6 | 0.309 | 210.1 |
> | Hand Force & Muscle EMG | 0.117 | 14.7 | 0.307 | 207.6 |
> | Hand Pose & Body Pose | 0.113 | 15.0 | 0.297 | 206.2 |
> | All sensory used | 0.110 | 16.0 | 0.276 | 203.5 |
>
> **Is the number of modalities the only thing that differs between unimodal and multmodal tests?**
> Yes. the number of modalities is the only thing that varies in the comparison to unimodal
>
> [1] Qi. et a. PointNet, CVPR 2015
>
> [2] Doimo, Diego, et al. "Redundant representations help generalization in wide neural networks." NeurIPS, 2022

---

> ### Author Response · Authors · 2024-11-20
> **Author Response (3/4)**
>
> **Computational efficiency scale with additional sensory inputs in compute time and memory usage**
>
> Thank you for this question, we have revised our manuscript to include computation efficiency *(parameter count) with respect to additional sensory inputs along with discussion on this additional advantage of our method, where each added sensory inputs only introduce very small computational coast compared to the video model backbone it self. Moreover, because the modalities can be processed in parallel, there is no additional computational time cost.
> We report the modules of our model in Sec. A 6.2 and Table 3 in the revised manuscript. Specifically, we can see from the table that the multimodal action signal module is fairly small compared to the video module. Each signal average to around 18044828 parameters which is only 5 percent of the total model weights. The lightweight action signal heads highlights the advantage of our method for low memory added for each action signal modality. As each signal modality is independent and can be processed in parallel, there is no addtional computational time cost for the added signal modality
>
>
> | Module                               | Parameter Count | Percentage of Total |
> |--------------------------------------|-----------------|---------------------|
> | Signal Expert Encoder for 4 modalities | 43,780,932      | 0.13                |
> | Signal Projection   for 4 modalities        | 11,537,408      | 0.03                |
> | Signal Decoder for 4 modalities      | 28,398,382      | 0.08                |
> | Signal Total   for 4 modalities                        | 83,716,722      | 0.25                |
> | Video Model                          | 252,380,168     | 0.75                |
> | Total Model                          | 336,096,890     | 1.00                |
>
> **The low-level policy optimization**
>
> We have also revised our manuscript Section 4 Downstream Application to hopefully make this more clear.
>
> The key difference between the baseline method and policy optimization with our proposed encoder is in the loss function. The task setup is similar to other existing goal conditioned policy optimization works [1, 2, 3], given starting state and goal state, predict the action trajectory. The baseline method is trained similar to other existing goal conditioned policy optimization works by directly regressing predicted action trajectory from the policy and the expert action trajectory. The only difference between policy trained with ours and the baseline method is the additional loss term that matches the simulated goal state from the predicted action using our proposed video simulator and the desired given the goal state.
>
> To better define the task setup in mathmatical terms, we compare training of the same policy network $p(\cdot){\pi{\theta}}$ under two conditions. The baseline method uses the commonly employed goal-conditioned policy training approach. This baseline is the policy network taking the starting state and target state, depicted by two video frames $x_0$ and $x_T$, and directly regress policy $\pi_{\theta}$ minimizing the L2 distance between the predicted action $ \hat{a}[1, T] = \pi_{\theta}(x_0, x_T) $ and ground truth expert action trajectory$a_{[1, T]}$.  This L2 loss term is defined as:  $ \mathcal{L}\_a = \sum\_{t} \| \hat{a}\_{t} - a\_{t} \|_2 = \| p(x\_0, x\_T)\_{\pi\_{\theta}} - a\_{[1, T]} \|_2 $.
>
> The second approach is to train the same policy $\pi_{\theta}$ in conjunction with our pretrained simulator. We feed the action trajectory predicted by the policy network:  $ \hat{a}\_{[1, T]} = \pi\_{\theta}(x_0, x_T) $ into our pretrained simulator model $g(\cdot)$ to predict the video frames from this action trajectory: $ \hat{x}\_T = g(p(x_0, x_T)\_{\pi\_{\theta}})\_{T}. $ This additional loss term is defined as: $ \mathcal{L}\_{sim} = \| \hat{x}\_T - x\_T \|\_2 = \| g(p(x_0, x_T)\_{\pi_{\theta}})\_{T} - x\_T \|_2. $ The total loss term for the second condition is $ \mathcal{L}\_{sim-policy} = \mathcal{L}\_a + \mathcal{L}\_{sim}. $
>
> We evaluate the effectiveness by using the L2 distance between the predicted action $\hat{a}\_{[1, T]}$ and the ground truth action $a_{[1, T]}$, which is defined as, $ \| \hat{a}\_{[1, T]} - a\_{[1, T]} \|_2. $
>
> - **How does the policy generate frames when the proposed simulator is removed**
> During training the baseline policy does not generate video frames. The goal of the policy is to predict multisensory action trajectory, given the starting state and target state.
>
> [1] Moritz Reuss. et al. Goal conditioned imitation learning using score-based diffusion policies. RSS, 2023.
>
> [2] Yiming Ding, et al. Goal-conditioned imitation learning. NeurIPS 32, 2019
>
> [3] Cheng Chi, et al. Diffusion policy: Visuomotor policy learning via action diffusion. The International Journal of Robotics Research, 2023.

---

> ### Author Response · Authors · 2024-11-20
> **Author Response (4/4)**
>
> **Adding the capability to predict multimodal sensory outputs**
>   - Thank you for the suggestion. Predicting multisensory action output can be achieved through optimizing a policy network, as shown in Sec. 4 in Downstream Application of our method. Experiments shown in downstream application section demonstrates prediction of multisensory outputs. The first experiment in Section 4 shows a low-level multisensory policy optimization example. We integrate our main method to help achieve more accurate multisensory action output.
>
>   - We would like to emphasize that the primary focus of our work is effective multisensory representation learning for the task of temporally fine-grained video simulation. Our task and method are designed to take multisensory action signals as input to predict future video sequences from context video frames and action signals. While our method can be extended to support policy learning for multisensory action trajectory prediction, as demonstrated in Section 4, our main contribution lies in advancing multisensory representation learning and understanding interoception-exteroception dynamics for video prediction.
>
>   - We would like to kindly invite the reviewer to clarify how the proposed suggestion of adding capability to predict multisensory output differs significantly from the experiments already presented in Section 4. We would be happy to address this concern and provide additional details.
>
>
> We hope that we addressed all your concerns. we would highly appreciate it if you could consider improving your score. If you would like us to conduct any additional experiments that are reasonable within the rebuttal period, kindly let us know, and we will try our best to accommodate your suggestions. If you are satisfied with our responses and changes. Please consider raising your mark and recommending our paper for acceptance.

---

> ### Author Response · Authors · 2024-11-22
>
> We would like to kindly invite to reviewer to review our responses to see if we addressed all their concerns. If you would like us to conduct any additional experiments that are reasonable within the rebuttal period, please let us know, and we will try our best to accommodate your suggestions. If you are satisfied with our responses and changes. Please consider raising your mark and recommending our paper for acceptance.

---

> ### Author Response · Authors · 2024-11-24
>
> Dear reviewer Xsn3,
>
> Thank you for your time and effort in reviewing our work.
>
> We would like to kindly invite you to take a look at our responses to see if we addressed all your concerns.
>
> If you would like us to conduct any additional experiments, we kindly ask you that you let us know sooner than later so we have enough time to run them.
>
> If you have further questions, please do not hesitate to ask.
>
> If you are satisfied with our responses and changes. Please consider raising your mark and recommending our paper for acceptance.
>
> Best,
>
> Submission 123 Authors

---

> ### Author Response · Authors · 2024-11-24
> **response to reviewer Xsn3**
>
> Dear reviewer Xsn3,
>
> Thank you for taking the time to review our responses and for your thoughtful reconsideration. We sincerely appreciate your feedback and are grateful that our clarifications could address your concerns.
>
> We have updated the Title in our manuscript. The aforementioned context to future multisensory action sequence and video frame generation sounds like a interesting direction. We hope our work can inspire future research in this direction.
>
> Thank you again for your thoughtful reconsideration. We sincerely appreciate your time and effort serving the community.

---

### Official Review · Reviewer_sUh3 · 2024-11-02

**Soundness:** 3
**Presentation:** 3
**Contribution:** 3
**Rating:** 6
**Confidence:** 4

**Summary:**

This paper presents a novel approach to multisensory simulation for fine motor control in general-purpose household robots. By introducing proprioception, kinesthesia, force haptics, and muscle activation, the authors aim to enhance generative video simulation models to better capture delicate control tasks. They propose a multimodal feature extraction paradigm that aligns sensory data to a shared representation space while preserving unique modality information. Additionally, a feature regularization method is introduced to enhance causality in action trajectory representations. The paper evaluates the model using the ActionSense dataset, demonstrating improvements in simulation accuracy and temporal consistency.

**Strengths:**

1. Originality: The introduction of a comprehensive set of interoceptive signals for generative simulation is novel, providing an innovative alternative to unimodal and text-based approaches in robotic control.
2. Quality: The method demonstrates clear improvements over baselines, especially in reducing temporal drift and enhancing fine-grained control, validated through detailed quantitative metrics and ablation studies.
3. Clarity: The experimental section is well-organized, allowing readers to understand the evaluation metrics and the comparative advantage of the proposed approach. Figures effectively illustrate the model’s performance across different sensory configurations.
This multisensory approach holds value for robotics, especially in applications requiring high-fidelity simulations for complex tasks, such as culinary or surgical activities, where precise control is essential.

**Weaknesses:**

1. The paper's reliance on a single dataset, ActionSense, restricts the method's demonstrated applicability. The dataset’s constraints—kitchen-based activities with a limited range of sensory inputs—limit the model's robustness across more varied settings. Extending the model's evaluation to diverse datasets involving different environments or robotic tasks would enhance its generalizability and applicability to real-world scenarios
2. Evaluation Scope: While the chosen metrics (MSE, PSNR, LPIPS, FVD) are appropriate for measuring video quality and temporal consistency, they do not address practical performance aspects such as the real-time execution capability on hardware-constrained robotic systems. Additionally, the experiments are limited to low-resolution (64x64) video output, which restricts the findings’ relevance for applications requiring high-fidelity visualizations. Testing on higher resolutions and considering hardware performance would strengthen the claim of practical applicability​.
3. Experimental Setup: The paper uses only subject five from ActionSense for testing while training on other subjects. This setup risks potential overfitting to specific motion patterns of a single subject, limiting cross-subject generalization. Cross-validation across subjects would help confirm the model’s robustness across various human motion characteristics​.

**Questions:**

1. Handling of Missing Sensory Modalities
Could the authors elaborate on the model's performance and stability when one or more sensory modalities are absent during testing? While the model’s robustness to missing modalities is briefly explored, further quantitative insights or analysis would strengthen the claim of adaptability. For real-world robotic applications, understanding how effectively the model compensates for missing inputs, such as through predictive mechanisms or alternative sensory weighting, would clarify its resilience to potential sensor failures.
2. Evaluation Beyond the ActionSense Dataset
The model’s evaluation is currently limited to ActionSense, which may not cover the full range of real-world tasks or interactions. Has any additional testing on varied datasets or environments been considered, or might it be feasible to cross-validate the model on similar multimodal datasets to confirm generalizability? Testing the model across broader contexts, such as more diverse daily activities or complex motor tasks, could reinforce its suitability for a wide range of applications and help mitigate dataset-specific limitations.
3. Computational Feasibility and Real-Time Constraints
Given the goal of deploying this approach in robotic applications, has any analysis been conducted on the model’s computational efficiency, especially in real-time or edge-computing scenarios? Insights on its processing requirements, latency, or suitability for deployment on constrained hardware would provide a clearer picture of its practicality. Specifically, understanding whether the model’s complexity scales with resolution or with the number of modalities would help determine the trade-offs between sensory input fidelity and processing efficiency.
4. Temporal Consistency and Model Drift Over Extended Simulations
While temporal drift reduction is highlighted as a strength, could the authors clarify the model’s performance stability in longer-term simulations or scenarios where prolonged fine motor control is needed? Given that robotic tasks often require sustained, accurate control, exploring how well the model mitigates accumulated drift over extended timeframes would enhance understanding of its practical performance in continuous operations.
5. Comparative Analysis with Multimodal Baselines
The paper mentions baseline comparisons with several multimodal feature extraction paradigms but does not provide a qualitative analysis of where the proposed model outperforms or falls short relative to these baselines. Would the authors consider offering a more detailed comparative breakdown, perhaps by examining specific cases where other methods fail or by isolating scenarios in which the proposed approach shows the most notable improvements? This comparison would be valuable in delineating the unique benefits of the model’s design choices.
Thank you for considering these questions and suggestions. I look forward to your responses, which I believe will further highlight the strengths and clarify the broader applicability of your approach！

---

> ### Author Response · Authors · 2024-11-20
> **Author Response (1/4)**
>
> We thank the reviewer for the detailed feedback and questions. We have made the following updates to our draft:
> 1. addtional cross-subject evaluation result
> 2. comprehensive ablation studies across all possible combinations of modalities,
> 3. additional experimental result analysis discussion comparing to the baseline methods.
> 4. computational efficiency model size table, showing that the proposed multimodal representation method is lightweight,
>
> and to our [supplementary website](https://sites.google.com/view/iclrsubmissionmultisensorysim):
> 1. higher resolution model
> 2. testing on other dataset, H2O
> 3. finetuning and testing on out of domain data
>
> we address individual questions below:
>
> **Higher resolution** We are training two higher-resolution models, one with a resolution of 128×128 and the other 192×192, matching the video resolution of existing generative video simulation paper, i.e. Unisim.  Preliminary qualitative predictions from these larger models are available on our   [supplementary website](https://sites.google.com/view/iclrsubmissionmultisensorysim).  We will continue to update the results as the models converge. The trained model weights will be made publicly available upon the acceptance of our paper.
>
>
> **Other Dataset** To show that our proposed method is generic is not designed for the ActionSense dataset, we conducted an experiment by directly applying our proposed approach on another dataset, H2O dataset [1]. H2O dataset is a unimodal action-video dataset that includes paired video and hand pose sequences. We would love to expand our our training on larger and more diverse dataset, However, to be best of our knowledge, ActionSense is the only dataset that includes paired multisensory action signal monitoring sequences alongside video sequences. We show experiment on H2O to demonstrates that our system is generic, not dataset specific, and can achieve reasonable performance when operating on other datasets .Qualitative results are provided on our  [supplementary website](https://sites.google.com/view/iclrsubmissionmultisensorysim). These results indicate that our model is capable of training and testing on unimodal action datasets, highlighting its generalizability beyond the ActionSense dataset. This demonstrates that our method is not specifically tailored to ActionSense and can adapt to various scenarios. We would love to expand our training on larger and more diverse datasets and bigger dataset featuring multisensory actions. Given that the primary contribution of our work lies in multisensory representation learning, these other datasets are not best suited to show the advantage of our work in fine-grained multisensory control. If there are other datasets featuring paired multisensory action signals and exteroceptive video data that we may have overlooked, we kindly invite reviewers to suggest them. We believe our proposed method offers a generalizable framework that can serve as a reference and can be applied more broadly as additional datasets of this nature become available.
>
> **Cross-subject evaluation** Thank you for your suggestion regarding additional cross-subject validation. Our choice to test and train on different subjects was specifically designed to demonstrate cross-subject generalization, and additional cross-subject validation can further illustrate this capability. To this end, we conduct two more cross-subject validation experiments, resulting in three in total. In these experiments, we validating on subject 2, subject 4, and subject 5, by training on the remaining four subjects, respectively. The results are included in Table 5 in Sec. A 6.3 in Appendix in our updated draft. We also investigated the slight variation in performance observed between Subject 2 and Subjects 4 and 5. It appears that Subject 2 occasionally demonstrates a slightly different manipulation style, such as using the palm and wrist area to hold potatoes in place while cutting. We suspect that this slight domain gap could come from variations in the fitness of sensor hardware, gender, or demographic differences.
>
> | Test subject    | MSE    | PSNR   | LPIPS   | FVD    |
> |-----------|---------|---------|----------|---------|
> | subject 2 | 0.115   | 15.8    | 0.301    | 206.7   |
> | subject 4 | 0.112   | 16.0    | 0.282    | 204.6   |
> | subject 5 | 0.110   | 16.0    | 0.276    | 203.5   |
>
> [1] Kwon, Taein  et al. H2O: Two Hands Manipulating Objects for First Person Interaction Recognition, ICCV 2021

---

> ### Author Response · Authors · 2024-11-20
> **Author Response (2/4)**
>
> **Video metrics do not address practical performance aspects such as the real-time execution capability on edge devices and robotic systems**
>
> - **Video metrics** Thank you for this question. We use video evaluation metrics because our paper focuses on accurate video generation through learning multisensory representation, as described in our problem statement in line 86-93. Therefore, our evaluation metrics focuses on measuring the simulated video quality.
>
> - **Real time execution and Edge devices for robotics**
>   - Real-time execution and edge device computing is an interesting aspect. We would like to highlight that our work proposes a multisensory conditioned video simulator. When employed in robotics applications, simulator are used in to train policy networks. Normally, only the trained policy network, rather than the simulator itself, needs to be deployed on edge devices / robots. In general, simulators, including ours, do not require to be executed on edge devices or robots for real-time deployment.
>
>   - We show such application in Sec. 4 Downstram application. Similar to UniSim or any other robotic simulators, we train a goal-conditioned policy network using our pretrained video model. We directly adopt diffusion policy [1] as our policy network, which is lightweight (shown below) and can be executed on Jetsons [2]. We have added this parameter count and model size for our implementation of the multisensory action diffusion policy network in the Table 4, Sec. A6.2 in Appendix.
>
> | Module | Parameter Count |  float16 in MB | float32 in MB |
> | --- | --- | --- | --- |
> | Diffusion Policy Network | 120690484 | 241MB  | 482 MB |
>
> - **Computational complexity and number of modalities** We have revised our manuscript to include computation efficiency (parameter count) along with discussion on the additional advantage of our method, where each added sensory inputs only introduce very small computational coast compared to the video model backbone it self.
>   We report the modules of our model in Sec. A 6.2 and Table 3 in the revised manuscript. Specifically, we can see from the table that the multimodal action signal module is fairly small compared to the video module. Each signal average to around 18044828 parameters which is only 5 percent of the total model weights. The lightweight action signal heads highlights the advantage of our method for low memory added for each action signal modality. As each signal modality is independent and can be processed in parallel, there is no addtional computational time cost for the added signal modality.
>
> | Module                               | Parameter Count | Percentage of Total |
> |--------------------------------------|-----------------|---------------------|
> | Signal Expert Encoder for 4 modalities | 43,780,932      | 0.13                |
> | Signal Projection   for 4 modalities        | 11,537,408      | 0.03                |
> | Signal Decoder for 4 modalities      | 28,398,382      | 0.08                |
> | Signal Total   for 4 modalities                        | 83,716,722      | 0.25                |
> | Video Model                          | 252,380,168     | 0.75                |
> | Total Model                          | 336,096,890     | 1.00                |
>
> We hope these clarifications address your concerns and further illustrate the efficiency and potential of our method.
>
> [1] Cheng Chi, et al. Diffusion policy: Visuomotor policy learning via action diffusion. The International Journal of Robotics Research, 2023.
>
> [2] NVIDIA Jetson: AI platform for edge computing. NVIDIA. [https://developer.nvidia.com/embedded/jetson-modules](https://developer.nvidia.com/embedded/jetson-modules)

---

> ### Author Response · Authors · 2024-11-21
> **Author Response (3/4)**
>
> **Analysis on Robustness to missing modalities**
>   - We provide a comprehensive experiment on test time missing modalities. Testing with one missing modality are already included in Table 1 (b), and  testing with one modality present can be found in Appendix Table 2 in original draft / Table 6 in new draft. In addition, we have prepared more experiments showing testing with all combinations of paired modalities, included in Table 7. Table. 7 along with Table. 6 and Table.1b makes a comprehensive ablation study encompassing all possible combinations of test-time missing modalities.
>
>   - To aid the understanding of robustness to missing modalities during test-time, we make an inaccurate analogy to PointNet[4]. There are many points in the point cloud, but only few points are critical to decide the shape category. PointNet is known for robustness, the max operator in PointNet selects these critical information for the entire point cloud. Similarly in our work, the cross-channel softmax selects most representative information for manipulation, *obscuring the source modality* that it came from, which is the main reason for the robustness towards missing modality.
>
>   - As discussed in Sec 3.3 in our manuscript, we can see from Table 1 (b), our model can achieve minimal performance drop when single modality is removed during test time, showing that our model trained on all modalities is robust to missing modalities during test time. We also notice a increasing trend of performance decrease as more and more modalities are remove during test-time, as shown in Table 6 ( originally 1) and Table 7 and in the Appendix section. Nonetheless, we can see that our model trained with multisensory action information still outperforms unimodal models, comparing Table in Figure 3 (a) , where the models are trained with unimodal information, and Appendix Table 2 / Table 6 in the updated draft, where the model trained with all sensory modalities and tested with one modality. This also shows the advantage in both robustness and performance to employ our method training on multimodal action signals.
>
>   - We hope this additional results on all possible combination of test-time missing modalities helps address your concern.
>
>
>
> **Temporal Drift and Error accumulation**
>   - Thank you for this question. We would like to point out that there are two sources to temporal drift / error accumulation in generative video simulation. One source is in the temporal fine-grained-ness of the action conditioning feature, and the other cause of error accumulation is the video model itself. We show in our work that leveraging temporally fine-grained multisensory action signals help significantly reduce temporal drift, when compared with language conditioning, unimodal conditioning, or other multimodal representation conditioning given the same video generative model backbone.
>
>   - The second source of temporal drift is inherent in the video model itself. Our work does innovate on this end of video model architecture, instead, we employ an existing video generative model, i2vgen [1]. However, because our proposed multimodal representation is generic, where various multimodal information is one latent feature, it is not specific to any video model architecture and can be used when more powerful video generative models become available.
>
>   - For robotic applications, when optimizing goal-conditioned policy networks, our method serves as an additional loss signal to help regularize regression of action trajectories, as shown in Sec. 4 in our paper. In the worst case, if the video model is untrained and predicts Gaussian noise given any input, the additional supervision signal provided by the video model will be ineffective, and the training of the policy network becomes entirely dictated by the regression of the action trajectory itself. In this form of application, when simulators are used to provide additional supervision signal to imitate high-dimensional multisensory expert trajectories, error-free simulation is preferred but not critical. For robotic tasks involving long-horizon planning, only using current generative video simulators, such as our work or unisim[2], might not be sufficient. Alternatively, one can employ high-level reasoning method such as Palm-E[3], to break down and localize tasks to be shorter horizon.
>
>   - Finally, we would like to further highlight the novelty of our work, 1 we introduce temporally fine-grained multisensory action signals to generative video simulation, 2. we propose an effective multisensory feature learning method. We show application but do not innovate or claim contribution on these other ends, i.e. video model architecture, robotic policies.
>
> [1] https://arxiv.org/abs/2311.04145
>
> [2] Yang, Mengjiao, et al. "Learning interactive real-world simulators." ICLR 2024.
>
> [3] Driess, Danny, et al. "Palm-e: An embodied multimodal language model." ICML 2023.
>
> [4] Qi et al. PointNet CVPR, 2015.

---

> ### Author Response · Authors · 2024-11-21
> **Author Response (4/4)**
>
> **Qualitative analysis over baseline approaches** Thank you for this comment. We have revised our draft to add more detailed experimental result analysis comparing our proposed approach and each baseline approaches. The added analysis can be found in Sec. 3.2. Specifically, ImageBind and Mutex aligns action signal modalities to the encoded video frames, where Imagebind uses contrastive loss to align action and visual features and Mutex uses L2 loss to directly regress the features between various modalities and the pretrained CLIP encoded visual feature. As very similar action motion trajectories can work with different visual contexts, matching action modality feature directly to various visual context creates a one-to-many mapping problem, making it difficult for the network to extract the intrinsic motion from the visual context, leading to significant error accumulation. Moreover, action signals and visual observation are modalities of large spatial disparity, directly regressing them leading to mode collapse when predicting future video frames. Similarily, Languagebind use contrastive loss to align various signal modalities to the encoded text descriptions. Constrastive losses magnify similarity between the participating features. Thus, training to match action sensory features to text features wipes out the temporal fine-grained information from the encoded action signals, leading to compromised predictions. Signal Agnostic Learning on the other hand does not use contrastive learning. By allowing gradient from different signal modalities to directly optimize the same latent manifold, Signal Agnostic approaches seem to outperform other baseline methods. However, these approaches induce loose coupling between the action signal modalities and the exteroceptive video modality, resulting in significant error accumulation.

---

> ### Author Response · Authors · 2024-11-22
>
> We would like to kindly invite to reviewer to review our responses to see if we addressed all their concerns. If you would like us to conduct any additional experiments that are reasonable within the rebuttal period, please let us know, and we will try our best to accommodate your suggestions. If you are satisfied with our responses and changes. Please consider raising your mark and recommending our paper for acceptance.

---

> ### Author Response · Authors · 2024-11-23
>
> Dear reviewer sUh3,
>
> Thank you for your time and effort in reviewing our work.
>
> We would like to kindly invite you to take a look at our responses to see if we addressed all your concerns.
>
> If you would like us to conduct any additional experiments that are reasonable within the rebuttal period, please let us know.
>
> If you have further questions, please do not hesitate to ask them.
>
> If you are satisfied with our responses and changes. Please consider raising your mark and recommending our paper for acceptance.
>
> Best,
>
> Submission 123 Authors

---

> ### Author Response · Authors · 2024-11-24
>
> Dear reviewer sUh3,
>
> Thank you for your time and effort in reviewing our work.
>
> We would like to kindly invite you to take a look at our responses to see if we addressed all your concerns.
>
> If you would like us to conduct any additional experiments, we kindly ask that you let us know sooner than later, so to leave us enough time to conduct them.
>
> If you have further questions, please do not hesitate to ask.
>
> If you are satisfied with our responses and changes. Please consider raising your score and recommending our paper for acceptance.
>
> Best,
>
> Submission 123 Authors

---

> ### Comment · Area_Chair_omip · 2024-11-25
>
> Dear Reviewer,
>
> Please provide feedback to the authors before the end of the discussion period, and in case of additional concerns, give them a chance to respond.
>
> Timeline: As a reminder, the review timeline is as follows:
>
> November 26: Last day for reviewers to ask questions to authors.
>
> November 27: Last day for authors to respond to reviewers.

---

> ### Author Response · Authors · 2024-11-25
>
> We would like to thank our AC for helping engage the reviewer.
>
> Dear reviewer sUh3,
>
> Thank you for your time and effort in reviewing our work.
>
> We would like to kindly invite you to take a look at our responses to see if we addressed all your concerns.
>
> Since Nov 26th is the last day for reviewers to ask questions, should you have any further questions, please do not hesitate to ask them.
>
> If you are satisfied with our responses and changes. Please consider raising your score and recommending our paper for acceptance.
>
> Best,
>
> Submission 123 Authors

---

> ### Author Response · Authors · 2024-11-26
> **Author Response (1/2) to Reviewer sUh3 comment "Further Feedback and Willingness to Reassess"**
>
> >Dear Submission 123 Authors,
> >Thank you for your detailed and thoughtful responses to the comments I raised earlier. I appreciate the extensive additional experiments, including the higher-resolution models, cross-subject evaluations, and robustness analysis to missing modalities. These efforts significantly improve the clarity, applicability, and robustness of your work.
> >Your research is innovative and impactful, and I recognize the meaningful contributions it makes to the field. I would be happy to reassess and raise my score further if you continue to refine key aspects, such as further expanding dataset testing or providing more detailed deployment analyses for real-world applications.
> >Please feel free to share additional updates or experiments during the rebuttal period. I am happy to provide further feedback if needed and look forward to seeing the finalized version of your promising work.
>
>
> Dear Reviewer sUh3,
>
> Thank you for recognizing our work as innovative and impactful, and also for confirming that you are willing to reassess our work with rebuttal. We address these newly requested experiments accordingly:
>
> **Further expanding dataset testing:**
>
> - We would first like to emphasize that we have already demonstrated the wide applicability of our proposed method by including an additional experiment in the rebuttal, where we tested our approach on a different dataset, the H2O [1] dataset, as outlined in our earlier response **(Author Response 1/4: Other Dataset)**.
> - We are willing to further expand testing of our proposed method on more additional datasets, if these experiments can further aid the audience’s understanding of our work.
> - Obtaining, parsing, training, and testing on a new dataset is non-trivial effort. As Nov 27th is the last day to upload revision, it might not be feasible to complete these newly requested experiments in one day for revision. However, as further expanded testing on more of these datasets carries similar message to the H2O experiment already conducted, they do not provide much additional information about our work, excluding them thus does not detract from the validity or impact of our work.
>
> **Providing more detailed deployment analyses for real-world application:**
>
> - Since the our main contribution is a multimodal representation for video simulation, which is to be used as a simulator to train policy network, only the trained policy network needs to be deployed for real-world application. We show the policy network [2] as employed in Sec. 4 downstream application in our manuscript. We use pytorch FlopCounterMode to count flops, and deployment analyses follow:
>
>     - throughput [3]
>         | Hardware type | NVIDIA Jetson Nano | Jetson Xavier | Jetson Orin NX | Jetson AGX Orin | RTX 4090 | H100 |
>         | --- | --- | --- | --- | --- | --- | --- |
>         | throughput (FPS) | 166 ~ 111 | 415 ~ 290 | 1,725 ~ 1,293 | 2,555 ~ 1,916 | 26,528 ~ 19,896 | 315,141 ~ 236,356 |
>     - latency (from ideal to assumed 70% hardware efficiency)
>         | Hardware type | NVIDIA Jetson Nano | Jetson Xavier | Jetson Orin NX | Jetson AGX Orin | RTX 4090 | H100 |
>         | --- | --- | --- | --- | --- | --- | --- |
>         | latency (in miliseconds) | 6.6 ms ~ 9.2 ms | 2.4 ms ~ 3.44 ms | 0.57 ms ~ 0.77 ms | 0.39 ms ~ 0.52 ms | 0.037 ms ~ 0.050 ms | 0.00317 ms ~ 0.00423ms |
>     - energy consumption
>
>         | Hardware type | NVIDIA Jetson Nano | Jetson Xavier | Jetson Orin NX | Jetson AGX Orin | RTX 4090 | H100 |
>         | --- | --- | --- | --- | --- | --- | --- |
>         | energy consumption (J) | 0.06 ~ 0.09 J | 0.036 ~ 0.051 J | 0.0114 ~ 0.0154 J | 0.0195 ~ 0.026 J | 0.01665 ~ 0.02250 J | 0.02219 ~ 0.03014 J |
>     - size / memory foot print of policy network
>         | Module | Parameter Count | float16 in MB | float32 in MB |
>         | --- | --- | --- | --- |
>         | Diffusion Policy | 120690484 | 241MB | 482 MB |
>
> - This experiment shows that a trained policy network can be feasibly deployed on edge hardware / more advance compute for real-world applications.
>
> - We would like to further clarify that the main focus of our work is **multisensory representation learning** for video simulation, we are positioned **not as a robotics paper**, nor do we claim to solve robotics directly. We appreciate that reviewers find our downstream applications interesting, we are happy to answer any questions regarding policy optimization experiments included in the paper. As robotics is not the main focus of our work, we hope to **not divert** attention from our main focus by developing additional experiments specifically in the robotics domain.
>
> Thank you again for your feedback. We hope these additional experiments help address your additional concerns for you to reassess our work and raise your rating to recommend us for acceptance.
>
> [1] Kwon, Taein et al. H2O, ICCV 2021
>
> [2] Cheng Chi, el. Diffusion policy. RSS, 2023.
>
> [3] https://developer.nvidia.com/embedded/jetson-modules

---

> ### Author Response · Authors · 2024-11-27
> **Author Response (2/2) to Reviewer sUh3 comment "Further Feedback and Willingness to Reassess"**
>
> > Dear Submission 123 Authors, Thank you for your detailed and thoughtful responses to the comments I raised earlier. I appreciate the extensive additional experiments, including the higher-resolution models, cross-subject evaluations, and robustness analysis to missing modalities. These efforts significantly improve the clarity, applicability, and robustness of your work. Your research is innovative and impactful, and I recognize the meaningful contributions it makes to the field. I would be happy to reassess and raise my score further if you continue to refine key aspects, such as further expanding dataset testing or providing more detailed deployment analyses for real-world applications. Please feel free to share additional updates or experiments during the rebuttal period. I am happy to provide further feedback if needed and look forward to seeing the finalized version of your promising work.
> --reviewer sUh3
>
> Dear Reviewer sUh3,
>
> We again thank you for confirming that you “would happy to reassess and raise your score, if we **further expanding dataset testing** or **providing more detailed deployment analyses for real-world applications.**” The results for the latter request is included in the above response.
>
> As per your first request, we have parsed **another dataset HoloAssist[1]**. Similar to the H2O dataset[3], it has paired hand pose and video. We have started to train our method and video model on this dataset, and we include this additional experiment on our [supplementary website](https://sites.google.com/view/iclrsubmissionmultisensorysim), and will continue to update results as the model is training and converging.
>
> We hope that **all the experiments on additional datasets**, ActionSense[2] (in original paper), H2O[3] (added during rebuttal), and HoloAssist[1] (added during rebuttal) can help address your additional concern on “**further expanding dataset testing**”. We kindly invite you to take a look at these additional experiments and response. We understand that further experiment requests are not advised during the extended rebuttal period, but we can answer any additional questions you may have about our work, especially about our main contribution.
>
> Again, thank you for confirming that our work is “innovative and impactful,” and recognizing “the meaningful contributions it makes to the field." If there is any concern holding you back from recommending our paper for acceptance, please do not hesitate to ask.
>
> We kindly ask that you reassess our work considering all rebuttal experiments, manuscript revision, and answers to questions. Since we have already conducted the additional experiments you requested and we were able to successfully address all your previous concerns, we hope that you can raise your score to recommend our paper for acceptance.
>
> Best,
>
> Submission 123 Authors.
>
> [1] Wang, Xin, et al. "Holoassist: an egocentric human interaction dataset for interactive ai assistants in the real world." ICCV 2023.
>
> [2] DelPreto, Joseph, et al. "ActionSense: A multimodal dataset and recording framework for human activities using wearable sensors in a kitchen environment." NeurIPS, 2022
>
> [3] Kwon, Taein et al. H2O: Two Hands Manipulating Objects for First Person Interaction Recognition, ICCV 2021

---

> ### Author Response · Authors · 2024-11-30
> **Author request for response per reviewer sUh3's comment "Further Feedback and Willingness to Reassess"**
>
> > Dear Submission 123 Authors, Thank you for your detailed and thoughtful responses to the comments I raised earlier. I appreciate the extensive additional experiments, including the higher-resolution models, cross-subject evaluations, and robustness analysis to missing modalities. These efforts significantly improve the clarity, applicability, and robustness of your work. Your research is innovative and impactful, and I recognize the meaningful contributions it makes to the field. I would be happy to reassess and raise my score further if you continue to refine key aspects, such as further expanding dataset testing or providing more detailed deployment analyses for real-world applications. Please feel free to share additional updates or experiments during the rebuttal period. I am happy to provide further feedback if needed and look forward to seeing the finalized version of your promising work. --reviewer sUh3
>
> Dear Reviewer sUh3,
>
> Thank you for recognizing our work as innovative, impactful, and makes meaningful contribution to the field.
>
> We would like to kindly invite you to take a look at the additional experiment we conducted upon your request. Thank you for your willingness to reassess our work. If you have questions regarding our work that's holding you back from recommending our paper for acceptance, please let us know.
>
> We hope that these additional experiments conducted per your request have addressed your additional concerns. If so, please consider raising your score and recommending our paper for acceptance.
>
> Best,
>
> Submission 123 Authors.

---

> ### Author Response · Authors · 2024-12-01
> **Author request for response per reviewer sUh3's comment "Further Feedback and Willingness to Reassess"**
>
> > Dear Submission 123 Authors, Thank you for your detailed and thoughtful responses to the comments I raised earlier. I appreciate the extensive additional experiments, including the higher-resolution models, cross-subject evaluations, and robustness analysis to missing modalities. These efforts significantly improve the clarity, applicability, and robustness of your work. Your research is innovative and impactful, and I recognize the meaningful contributions it makes to the field. I would be happy to reassess and raise my score further if you continue to refine key aspects, such as further expanding dataset testing or providing more detailed deployment analyses for real-world applications. Please feel free to share additional updates or experiments during the rebuttal period. I am happy to provide further feedback if needed and look forward to seeing the finalized version of your promising work. --reviewer sUh3
>
>
> Dear Reviewer sUh3,
>
> Thank you for recognizing our work as innovative, impactful, and that it makes meaningful contribution to the field.
>
> Thank you for your willingness to reassess our work. We would like to kindly invite you to see the additional experiment we conducted upon your request. If you have questions regarding our paper that's holding you back from recommending our paper for acceptance, please let us know.
>
> We hope that these additional experiments conducted per your request have addressed your additional concerns. If so, please consider raising your score and recommending our paper for acceptance.
>
> Best,
>
> Submission 123 Authors.

---

### Official Review · Reviewer_8rH5 · 2024-11-04

**Soundness:** 3
**Presentation:** 3
**Contribution:** 3
**Rating:** 6
**Confidence:** 3

**Summary:**

This paper presents a method for improving generative simulations in household robots incorporating multisensory inputs such as proprioception, kinesthesia, force haptics, and muscle activation. These sensory signals support delicate, real-time control essential for tasks involving fine motor skills.

The authors introduce a feature learning paradigm that aligns sensory modalities in a shared representation space while retaining each modality's unique contributions. A regularization scheme enhances causality in action trajectories, allowing for a more accurate representation of interaction dynamics.

The model achieves improvements in simulation accuracy and reduced temporal drift, outperforming baseline models. Comprehensive experiments and suggested applications, such as policy optimization and planning, demonstrate the model’s effectiveness in downstream applications

**Strengths:**

1. This work integrates multimodal information into generative simulations (video generation), addressing a reasonably novel task. The proposed pipeline effectively tackles multimodal misalignment using cross-attention, channel-wise softmax, and relaxed hyperplane interaction.

2. Extensive experiments show reasonable improvements over the previous state-of-the-art method, UniSim, across four metrics (MSE, PSNR, LPIPS, and FVD), with consistent improvements across different prediction time spans.

3. The ablation study confirms the effectiveness of using various modality data and validates different design choices. Interesting findings include a high correlation between hand force and muscle EMG data.

4. Clear visualizations in both the main paper and appendix enhance understanding of the method and results.

**Weaknesses:**

1. The model is trained solely on the ActionSense dataset, which limits the generalizability of the results.

2. The experiments are conducted with 64x64 resolution videos, which may be too low for real-world applications, such as robotic manipulation.

3. Some experiments are incomplete: in Table 3, the "Ours with sentence" result is missing, which would strengthen the study.

**Questions:**

1. Since in Table 1. there is some result in missing some of the domain data, could the proposed method apply to out-of-domain videos without multimodal information, such as the EpicKitchen dataset?

---

> ### Author Response · Authors · 2024-11-20
> **Author Response (1/2)**
>
> We thank the reviewer for the constructive comments. We have updated our manuscript with:
> 1. results on other dataset,
> 2.  experiment with out-of-distribution data,
> 3.  'ours with sentence' result, and
> 4. preliminary results of higher resolution model [supplementary website](https://sites.google.com/view/iclrsubmissionmultisensorysim).
>
> Higher resolution video model of 192 x 192 and 128 x 128 are continued to be updated as the models converge.
>
> **Other Dataset** To show that our proposed method is generic is not designed for the ActionSense dataset, we conducted an experiment by directly applying our proposed approach on another dataset, H2O dataset [1]. H2O dataset is a unimodal action-video dataset that includes paired video and hand pose sequences. We would love to expand our our training on larger and more diverse dataset, However, to be best of our knowledge, ActionSense is the only dataset that includes paired multisensory action signal monitoring sequences alongside video sequences. We show experiment on H2O to demonstrates that our system is generic, not dataset specific, and can achieve reasonable performance when operating on other datasets .Qualitative results are provided on our  [supplementary website](https://sites.google.com/view/iclrsubmissionmultisensorysim). These results indicate that our model is capable of training and testing on unimodal action datasets, highlighting its generalizability beyond the ActionSense dataset. This demonstrates that our method is not specifically tailored to ActionSense and can adapt to various scenarios. We would love to expand our training on larger and more diverse datasets and bigger dataset featuring multisensory actions. Given that the primary contribution of our work lies in multisensory representation learning, these other datasets are not best suited to show the advantage of our work in fine-grained multisensory control. If there are other datasets featuring paired multisensory action signals and exteroceptive video data that we may have overlooked, we kindly invite reviewers to suggest them. We believe our proposed method offers a generalizable framework that can serve as a reference and can be applied more broadly as additional datasets of this nature become available.
>
>   - **Can the method be applied to non multimodal (Unimodal Action) Data** Yes, our method can be directly applied on Unimodal action data, and thank you for suggesting the EpicKitchen dataset [2]. While this dataset offers bounding boxes for hand-object detection, it does not include action sensory signals such as hand pose or force. We conduct an experiment such on the H2O dataset discussed above. However, we emphasize that the primary focus and contribution of our work lie in addressing heterogeneous sensory modalities. Testing on single modality dataset shifts attention away from this core contribution. If there are other datasets featuring paired multisensory action signals and video data that we may have overlooked, we kindly invite reviewers to suggest them. We believe our proposed method offers a generalizable framework that can be applied more broadly as additional datasets of this nature become available.
>
> - **Dealing with out-of-domain (OOD) data** We present a second experiment to demonstrate that our method can handle specific out-of-distribution (OOD) scenarios through fine-tuning. For this experiment, we modified the original ActionSense dataset to create OOD data. Using LangSAM, we extracted segmentation masks for "potatoes" and recolored them to appear as "tomatoes." Since the video model had not encountered red vegetables or fruits during training, we fine-tuned our pretrained model on a small dataset of approximately 600 frames (30 seconds) and evaluated it on the test split of this "tomato" data. The data creation procedure and results on this experiment can also be found on our [supplementary website](https://sites.google.com/view/iclrsubmissionmultisensorysim) . The results show that the model achieves reasonable performance after fine-tuning. While we acknowledge that robust in-the-wild generalization requires training on larger-scale datasets with diverse domain coverage, this experiment illustrates a practical use case for addressing OOD data. Specifically, it demonstrates that by collecting a small, specialized dataset, our pretrained model can be effectively fine-tuned to adapt to new domains.
>
> We hope these experiments demonstrate the generalizability of our approach and its adaptability to different scenarios. Thank you for your thoughtful feedback, which has allowed us to further highlight the flexibility and contributions of our work.
>
> [1] Kwon, Taein  et al. H2O: Two Hands Manipulating Objects for First Person Interaction Recognition, ICCV 2021
>
> [2] Damen, Dima, et al. "Scaling egocentric vision: The epic-kitchens dataset."  ECCV 2018

---

> ### Author Response · Authors · 2024-11-20
> **Author Response (2/2)**
>
> **Higher Resolution Model**  We are training two higher-resolution models, one with a resolution of 128×128 and the other 192×192, matching the video resolution of existing generative video simulation paper, i.e. Unisim [1].  Preliminary qualitative predictions from these larger models are available on our   [supplementary website](https://sites.google.com/view/iclrsubmissionmultisensorysim).  We will continue to update the results as the models converge. The trained model weights will be made publicly available upon the acceptance of our paper.
>
>
>
> **"Ours with sentence"  result**
> Thank you for this question. We have updated our manuscript in Section 3.1, Table 1 (a) to include this results. We can see from the table that “ours with sentence” achieves similar performance to “ours with phrase.” This is because the sentences are generated from the action phrases based on sampling rate of frames and thresholding the hand force signal, as described in Appendix Sec A6.5 line 925-930. The original ActionSense dataset only offers action phrases. The sentences we generated do not introduce additional information beyond what is already captured by the temporally fine-grained multisensory action signals utilized by our method. For this reason, we initially omitted this experiment. However, we appreciate your suggestion, and we believe the inclusion of these results further clarifies our approach.
>
> | Model                 | MSE |  PSNR |  LPIPS |  FVD |
> |-----------------------|----------|----------|----------|----------|
> | Ours multisensory     | 0.110    | 16.0     | 0.276    | 203.5    |
> | Ours w/ phrase        | 0.113    | 16.0     | 0.274    | 200.4    |
> | Ours w/ sentence      | 0.111    | 16.0     | 0.274    | 201.7    |
>
>
>
> We hope that we fully answered your questions and addressed your concerns with our responses and changes to the manuscript. We thank you once again for your positive evaluation and hope that you will consider increasing your rating of our paper.
>
> [1] Yang, Mengjiao, et al. "Learning interactive real-world simulators." ICLR 2024.

---

> ### Comment · Area_Chair_omip · 2024-11-25
>
> Dear Reviewer,
>
> Please provide feedback to the authors before the end of the discussion period, and in case of additional concerns, give them a chance to respond.
>
> Timeline: As a reminder, the review timeline is as follows:
>
> November 26: Last day for reviewers to ask questions to authors.
>
> November 27: Last day for authors to respond to reviewers.

---

> ### Author Response · Authors · 2024-11-25
>
> We would like to thank our AC for helping engage the reviewer.
>
> Dear reviewer 8rH5,
>
> Thank you for your time and effort in reviewing our work. Your service to the community is much appreciated.
>
> If you have any further questions, please do not hesitate to ask them.
>
> We hope that we fully answered your questions and addressed your concerns with our responses and changes to our manuscript.
>
> We thank you once again for your positive evaluation and hope that you will consider increasing your rating of our paper.
>
> Best,
>
> Submission 123 Authors

---

> > ### Author Response · Authors · 2024-12-01
> >
> > Dear reviewer 8rH5,
> >
> > Thank you for your time and effort in reviewing our work. Your service to the community is much appreciated.
> >
> > If you have any further questions, please do not hesitate to ask them.
> >
> > We hope that we fully answered your questions and addressed your concerns with our responses and changes to our manuscript and supplementary website.
> >
> > We thank you once again for your positive evaluation and hope that you will consider increasing your rating of our paper.
> >
> > Best,
> >
> > Submission 123 Authors

---

> > > ### Comment · Reviewer_8rH5 · 2024-12-03
> > >
> > > Thank you for addressing my concerns in text and with additional experiments. I would like to keep my positive evaluations.

---

### Author Response · Authors · 2024-11-20
**General Comment by Author**

We thank all reviewers and ACs for their time and effort in reviewing our paper and for their constructive feedback and we are glad that the reviewers find the following contributions of our work:

**Novelty**: This work introduces a comprehensive set of interoceptive signals for generative simulation is novel (8rH5, sUh3, Xsn3, sUh3) and is well motivated (sUh3, qMjQ).

**Method**: The proposed approach is effective (8rH5, sUh3, qMjQ, Xsn3). Useful for downstream robotic applications (h9Bd).

**Experiments**: The work conducts thorough and concrete experiments and evaluations, showing consistent improvements (8rH5,sUh3,  Xsn3, qMjQ). Ablation studies confirm effectiveness and offer insights (8rH5, Xsn3).

**Presentation**: The paper is well organized and well explained (sUh3, qMjQ) accompanied with clear visualization of results and figures (8rH5, sUh3, qMjQ)

**Usefulness**: This work offers a useful system for downstream applications that requires precise control and fine-grained motor movements (sUh3, Xsn3) and can inspire future research (qMjQ).

---

We will address reviewers' concerns in the individual responses. We also revised our manuscript according to the reviewers' suggestions, and we would like to note that we made the following **major updates to our manuscript** and [supplementary website](https://sites.google.com/view/iclrsubmissionmultisensorysim) following the suggestions by the reviewers (revision shown in blue in the updated manuscript):

- Added Experiment:
    - Higher Resolution Model (Supplementary Website) — (8rH5, sUh3)
    - Testing on other Dataset — H2O Handpose Data  (Supplementary Website) -- (8rH5, sUh3, Xsn3)
    - Testing on more other Dataset — HoloAssist Handpose Data  (Supplementary Website) -- (sUh3)
    - Testing on out of domain data (Supplementary Website) — (qMjQ)
    - Results on Ours with Sentence — (Section 3.1 Table 1.(a)) —(8rH5)
    - Ablation studies for across all possible combinations of modalities -- (Xsn3)
    - Cross-subject evaluation (Sec. A 6.3) -- (sUh3)
- Added Discussion and Clarification :
    - Clarification on Notation (Sec. 2, Sec. A6.1) — (Xsn3)
    - Detail Analysis on comparing each multimodal representation learning baselines (Sec. 3.2) — (sUh3)
    - Additional Pipeline Figure and Added arrow to original method figure — (Fig.1 Sec 2, Fig 12 in Sec 6.1) -- (Xsn3)
    - Model Size break down (Sec. A5) -- (sUh3)
    - Clarification on Low-level policy optimization experiment setup (Sec. 4) — (Xsn3)
    - Comparison between training and testing with ablated modalities — (Sec. 3.3) — (Xsn3)

In addition to submitting a revised paper, we reply to each reviewer individually to address their questions and concerns and address some general concerns below. We hope that these responses together with the revised manuscript clear up any confusion and resolve all issues that the reviewers had, and that you will consider increasing your rating for our paper.

---

> ### Author Response · Authors · 2024-11-20
> **General Comment by Author Addressing Common Questions**
>
> **Higher resolution model (8rH5,sUh3, Xsn3)** We are training two higher-resolution models, one with a resolution of 128×128 and the other 192×192, matching the video resolution of existing generative video simulation paper, i.e. Unisim.  Preliminary qualitative predictions from these larger models are available on our [supplementary website](https://sites.google.com/view/iclrsubmissionmultisensorysim).  We will continue to update the results as the models converge. The trained model weights will be made publicly available upon the acceptance of our paper.
>
> **Other Dataset (8rH5,sUh3)** To show that our proposed method is generic is not designed for the ActionSense dataset, we conducted an experiment by directly applying our proposed approach on another dataset, H2O dataset [1]. H2O dataset is a unimodal action-video dataset that includes paired video and hand pose sequences. We would love to expand our our training on larger and more diverse dataset, However, to be best of our knowledge, ActionSense is the only dataset that includes paired multisensory action signal monitoring sequences alongside video sequences. We show experiment on H2O to demonstrates that our system is generic, not dataset specific, and can achieve reasonable performance when operating on other datasets .Qualitative results are provided on our  [supplementary website](https://sites.google.com/view/iclrsubmissionmultisensorysim). These results indicate that our model is capable of training and testing on unimodal action datasets, highlighting its generalizability beyond the ActionSense dataset. This demonstrates that our method is not specifically tailored to ActionSense and can adapt to various scenarios. We would love to expand our training on larger and more diverse datasets and bigger dataset featuring multisensory actions. Given that the primary contribution of our work lies in multisensory representation learning, these other datasets are not best suited to show the advantage of our work in fine-grained multisensory control. If there are other datasets featuring paired multisensory action signals and exteroceptive video data that we may have overlooked, we kindly invite reviewers to suggest them. We believe our proposed method offers a generalizable framework that can serve as a reference and can be applied more broadly as additional datasets of this nature become available.
>
> **Edge Computing and Real Time Execution for Robotics (sUh3, Xsn3)**
>   - We would like to highlight that our work proposes a multisensory conditioned video simulator. When employed in robotics applications, simulator are used in to train policy networks. Normally, only the **trained policy network, rather than the simulator** itself, needs to be deployed on edge devices / robot hardware. In general, simulators, including ours, do not require to be executed on edge devices or robots for real-time deployment.
>
>   - We show such application in Sec. 4 downstream application. Similar to UniSim or any other robotic simulators, we train a goal-conditioned policy network using our pretrained video model. We directly adopt diffusion policy [1] as our policy network, which is lightweight show in Sec. A5 and can be easily executed on NVIDIA Jetsons.
>
>   - The main focus of our work is multisensory representation learning. While we show downstream applications in policy training, where our work facilitates policy optimization, we do not directly tackle or claim to solve robotics. We appreciate that reviewers find our downstream applications interesting, we kindly ask reviewers to focus on the main contribution of our work--multimodal representation.
>
> [1] Kwon, Taein et al. H2O: Two Hands Manipulating Objects for First Person Interaction Recognition,ICCV 2021
>
> [2] https://github.com/luca-medeiros/lang-segment-anything

---

### Author Response · Authors · 2024-11-20
**Message to AC and Reviewer**

Dear reviewers and AC,

We would like to thank you for your review and your service to the community. Your opinions are much valued, and we would like to see if we have successfully addressed your concerns with our rebuttal, and/or if you have additional concerns about our work.

We would like to invite reviewer sUh3, Xsn3 to raise any additional concerns that they might have. If you have additional questions/experiments that you'd like us to conduct, we kindly ask you to let us know soon than later, so to leave us enough time to run experiments and address your concerns. If we have addressed all your concerns, please consider reassess our work with the rebuttal experiments and clarification.

Again, thank you for your time and effort towards reviewing our paper. Your service to the community is much appreciated.

Best,

Submission 123 Authors

---

### Meta-Review · Area_Chair_omip · 2024-12-19

**Metareview:**

While the paper introduces a promising approach to multisensory-conditioned video simulation and received positive feedback from the reviewers, it unfortunately violates the strict page limit set by the conference. The final revised submission exceeds the 10-page limit by extending onto a 12th page, which strongly violates the guidelines, and might have impacted the discussion process. Another minor comment about the negligence of the authors, is that while they changed the title in the resubmitted paper, they did not update it in the openreview system.

**Additional Comments On Reviewer Discussion:**

During the rebuttal period, reviewers raised concerns about dataset generalizability, methodological clarity, robustness to missing modalities, and computational feasibility. The authors addressed these issues through additional experiments on new datasets, cross-subject evaluations, detailed clarifications, and computational analyses. However, the submission violated the strict 10-page limit, potentially influencing reviewer scores and undermining the fairness of the evaluation process. Despite the authors’ improvements, this violation cannot be overlooked, leading to paper rejection.

---

### Decision · Program_Chairs · 2025-01-22

Reject